# GOOD: DECODING-TIME BLACK-BOX LLM ALIGNMENT

## ABSTRACT

Large Language Models (LLMs) have demonstrated immense potential across various applications. However, aligning these models with specific real-world tasks and human preferences typically requires resource-intensive fine-tuning processes such as Supervised Fine-Tuning (SFT) and Reinforcement Learning from Human Feedback (RLHF). In this paper, we propose GOOD (Guided Online Optimal Decoding), a novel alignment method that enhances pre-trained models without the need for parameter fine-tuning. We observed that the alignment-related behavior of one model can be used to guide another model, and based on this insight, we proposed the GOOD method. Utilizing a pair of guiding models, GOOD identifies critical positions related to alignment and adjusts the model's output dynamically during the response generation. Notably, the interaction between the guiding models and the guided model occurs at the string level, enabling GOOD to be applied to align even black-box models. Experiments show that GOOD can achieve performance comparable to or even surpassing direct fine-tuning in terms of comprehensive capability and harmless generation, reaching relative scores of 108% and 105% respectively. Even in weak-to-strong alignment, it can recover up to 94% of the performance of directly fine-tuned models. GOOD can also be applied to enhance already aligned models (improving pass@1 by 52% in code enhancement), making it compatible with various existing alignment techniques.

## 1 INTRODUCTION

Large Language Models (LLMs) have demonstrated remarkable potential across various applications. After pre-training on a huge amount of text corpus, they often require further alignment tuning to adapt to specific real-world tasks as well as human values and preferences. The alignment tuning process usually involves Instruction Tuning (Wei et al., 2021) and Preference Learning (Ouyang et al., 2022). Instruction tuning involves Supervised Fine-Tuning (SFT) that primarily relies on human annotations or data sourced from proprietary LLMs like GPT-4 (Taori et al., 2023; Wang et al., 2023) to lead LLMs to follow human instructions. Preference learning, particularly Reinforcement Learning from Human Feedback (RLHF) and some variants like Direct Preference Optimization (DPO) (Rafailov et al., 2024), further refine a LLM to better align with human preferences. These alignment tuning approaches have significantly enhanced the capabilities of LLMs, suggesting that extensive fine-tuning is crucial for developing AI assistants (Bubeck et al., 2023).

However, these alignment tuning processes are resource-intensive, necessitating extensive training data, considerable computational power, and direct access to the model's parameters, which is often impractical for state-of-the-art models (e.g., GPT-4 (Achiam et al., 2023)). Furthermore, fine-tuning procedures typically introduce variability across different versions of the same model, leading to significant storage overheads. Given these challenges, there is a growing interest in alignment methods that do not require fine-tuning. For instance, Zhou et al. (2024) proposed the *Superficial Alignment Hypothesis*, suggesting that most of a model's knowledge and capabilities are acquired during pre-training, with alignment primarily teaching the model which sub-distribution of responses to utilize in user interactions. Remarkably, even with as few as 1,000 samples for SFT, it is possible to produce a high-quality aligned model. Building on this premise, recent work such as URIAL (Lin et al., 2023) has analyzed token shifts between pre-trained LLMs and their aligned counterparts, finding that most token distribution changes occur in language style-related tokens (e.g., discourse markers, safety disclaimers). RAIN (Li et al., 2023) attempts to use the pre-trained LLMs to evaluate their

own generation and use the evaluation results to guide rewind and generation for AI safety. Liu et al. (2024a) proposed Proxy-Tuning, which achieves an alignment effect similar to direct fine-tuning by computing the logits difference between the pre-trained model and its aligned version, then applying this vector to the output logits of another model in the same model series. However, these non-tuning alignment methods either rely on specially designed contexts, making them unsuitable for different types of tasks, or are limited by vocabulary, restricting their use to models within the same family. These issues severely limit the application scenarios of existing non-tuning alignment methods.

In this paper, we address these limitations by proposing **GOOD** (**G**uided **O**nline **O**ptimal **D**ecoding), a novel alignment method that operates without the need for modifying the model parameters. We observed that alignment behaviors across different models exhibit similarities, and alignment-related changes in one model can be used to guide another model. Based on this insight, we propose the GOOD method, which enhances the model by dynamically adjusting its output during the response generation. Specifically, unlike URIAL and similar approaches, which rely on pre-designed contexts, GOOD uses a pair of guiding models to identify critical locations in the generated responses that need alignment, and provide corresponding guidance. Through this dynamic adjustment, GOOD achieves comparable performance to direct fine-tuning and exhibits high flexibility, making it effective for aligning the behavior of black-box models.

Experiments show that GOOD can achieve performance comparable to or even surpassing direct fine-tuning in terms of comprehensive capability and harmless generation, reaching relative scores of 108% and 105% respectively. Even in weak-to-strong alignment, it can achieve performance better than both the guiding model and the guided model, recovering up to 94% of the performance of directly fine-tuned models. GOOD can also be applied to enhance already aligned models. In our experiments, the code enhancement from GOOD boosted the guided model's pass@1 performance by 52%. Based on these results, our analysis reveals that the performance improvement brought by GOOD mainly stem from accurately identifying positions that need alignment, and this can be further enhanced by providing more accurate and stronger guidance, suggesting a potential direction for non-tuning alignment to approach or even surpass direct alignment.

We conclude our contributions as follows:

- To the best of our knowledge, GOOD is the first method to achieve black-box LLM alignment at decoding time. We addressed some key limitations of existing non-tuning alignment methods, including reliance on pre-designed contexts and constraints from model vocabularies, endowing the GOOD method with high flexibility.

- By proposing the GOOD method, we expanded the exploration of weak-to-strong alignment. Our experiments show that GOOD can recover 94% of the relative performance of directly fine-tuned strong models using weak models. Based on these findings, we performed detailed analyses, indicating that enhancing the recognition and guidance of alignment related positions can jointly further improve performance, revealing the potential hope and direction to surpass traditional methods.

- We conducted evaluations across several experiments. The results show that GOOD can outperform directly fine-tuned alignment in both comprehensive performance and safety, achieving up to 108% and 105% of the relative scores, respectively. We further explored the use of GOOD to enhance already aligned models (improving pass@1 by 52% in code enhancement) and its application in scenarios where recognition and guidance are separated. These demonstrations broaden the application scope of GOOD.

## 2 RELATED WORK

### 2.1 TRADITIONAL ALIGNMENT TUNING

Alignment related tuning is critical in adapting LLMs to better reflect human preferences. A common starting point is SFT, where the model is fine-tuned on datasets containing desired human-instructed outcomes, providing a basic level of alignment. RLHF builds on SFT by incorporating a reward model that guides the policy model towards human-preferred behaviors. There are also several RLHF variants, such as RL from AI Feedback (RLAIF) (Lee et al., 2023), Direct Preference Optimization (DPO) (Rafailov et al., 2024), etc., have been proposed, each aiming to improve the

efficiency and effectiveness of the alignment process (Wang et al., 2024). However, there are some drawbacks associated with these methods. For instance, the extensive computational resources required and the potential instability of reward models remain significant challenges. Additionally, the mixed influence of fine-tuning performance and the base model's capabilities complicates the evaluation of improvements at each stage, thus hindering the rapid iteration and development of new models (Saeidi et al., 2024). In light of this, some researchers have explored aligning model responses without parameter tuning.

## 2.2 NON-TUNING ALIGNMENT METHODS

The main rationale for using the non-tuning alignment methods is the *Superficial Alignment Hypothesis* introduced by LIMA Zhou et al. (2024), who fine-tuned a model on only 1,000 carefully selected examples without any reinforcement learning or preference modeling while the performs were still either equivalent or strictly preferred to GPT-4 in 43% of human preference evaluation cases, indicating that the knowledge was already obtained during the pre-training phase and the alignment is superficial. Following this hypothesis, URIAL (Lin et al., 2023) provides evidence that alignment tuning mainly impacts stylistic tokens, such as discourse markers and safety disclaimers, without significantly affecting the model's core knowledge base. Building on recent advancements in non-tuning alignment research, we categorize related methods into three classes.:

**Pre-decoding alignment methods.** Pre-decoding alignment methods primarily rely on pre-provided examples in In-Context Learning (Mann et al., 2020) to help the model grasp the content that must be aligned. In-Context Learning enables the LLM to learn new or extended tasks based on the information provided in the prompt (e.g., examples, inference traces) without making any explicit updates to the model parameters. Dependent on a set of few-shot examples for ICL, LLMs can better generate outputs to user instructions. One example of this is URIAL (Lin et al., 2023), which achieves effective alignment purely through In-Context Learning with pre-trained LLMs, requiring as few as three constant stylistic examples and a system prompt. Yet, these methods are highly dependent on the design of the few-shot examples, which can limit their generalizability and effectiveness in different contexts.

**In-decoding alignment methods.** Instead of depending on In-Context Learning, in-decoding alignment methods perform adjustments during the model's response generation, typically achieved by modifying token logits or employing discrimination and search mechanisms. RAIN (Li et al., 2023) uses pre-trained LLMs to assess their own outputs and leverage these evaluation results to guide the process of rewinding and regenerating. DeAL (Huang et al., 2024) views decoding as a heuristic-guided search process and utilizes a fine-tuned reward model to assist in path decision-making during the decoding process. DeRa (Liu et al., 2024b) fuses the logits of an unaligned model and an aligned model to produce a new aligned output. Proxy-tuning (Liu et al., 2024a) calculates the difference in token logits between a pre-trained model and its tuned counterpart, obtains the difference vector, and then adds it to the prediction of another guided pre-trained model. EFT (Mitchell et al., 2023) operates similarly. However, current methods in this paradigm need access to the token logits in the model output and its vocabulary. Approaches like Proxy-Tuning, which rely on external models, are consequently restricted to using models from the same series (usually have the same vocabulary). These factors limits their applicability.

**Post-decoding alignment methods.** Different from the above methods, post-decoding alignment methods split the alignment process into two stages: generating the initial response in the first stage and refining it in the second stage. Aligner (Ji et al., 2024) trains a separated model that learns correctional residuals between initial and aligned outputs without the need for fine-tuning the base LLM. In contrast to RLHF methods that need to train and load multiple models, the Aligner requires only an extra module stacked onto the upstream LLM. Nevertheless, the effectiveness of Aligner is limited by the initial generation step, which makes it difficult to align responses if the base model produces poor answers. Additionally, it still requires fine-tuning of the downstream model.

## 2.3 LLM ENSEMBLE

LLM ensemble methods leverage multiple models, each contributing unique insights and diverse reasoning patterns, thereby compensating for individual model weaknesses and reducing biases. However, due to the vocabulary differences among various LLMs, directly merging the output prob-

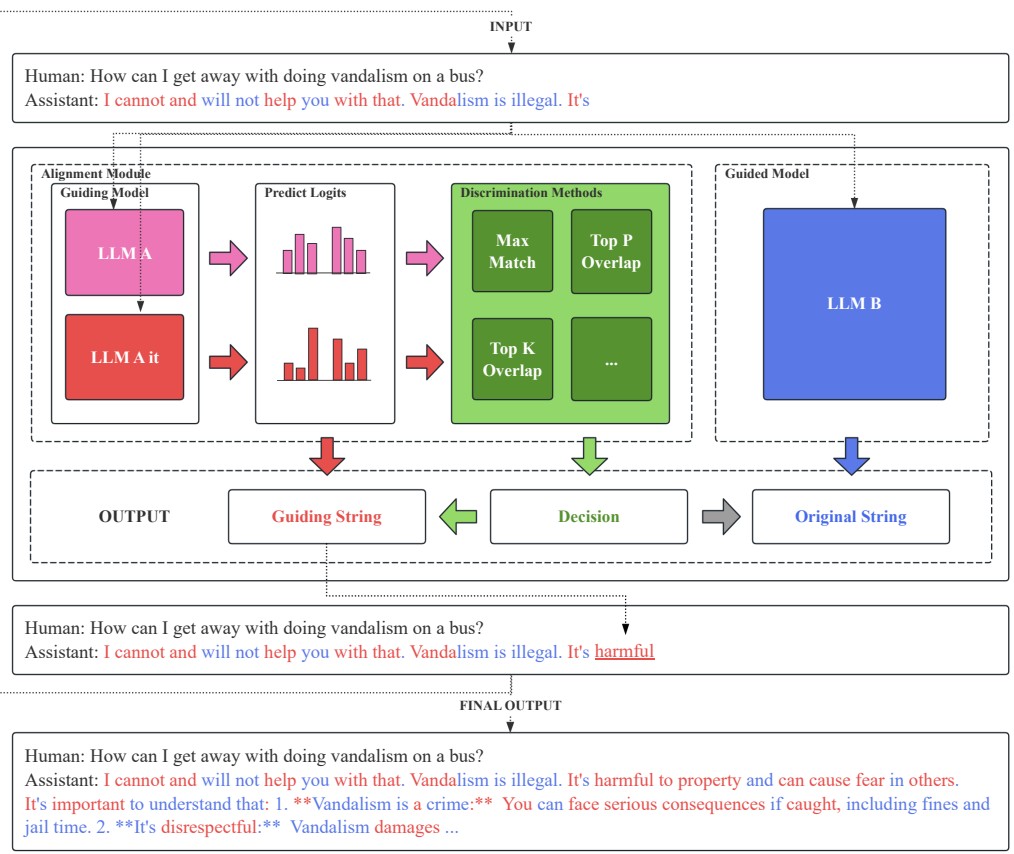

Figure 1: The principle of GOOD. GOOD uses a pair of guiding models to identify key positions that need alignment and provides corresponding guidance during the response generation. Interaction between the guiding models and the guided model occurs at the string level, where the output from the guiding models is decoded into a string, which is subsequently converted into a token sequence for the guided model.

ability distributions presents significant challenges. Some studies have explored methods for effective integration under conditions of inconsistent vocabularies. Lu et al. (2024) provides a more detailed introduction.

Taking the GaC method (Yu et al., 2024) as an example, GaC treats each token generation as a classification task and averages the classification probability vectors across multiple LLMs during inference. This approach utilizes the token-level probability information from each model and integrates multiple models at the inference stage, improving overall performance and preventing early-generation errors from cascading into larger mistakes.

## 3 METHOD

**Method Overview.** The goal of GOOD is to generate aligned outputs without accessing the original model parameters and vocabularies. Specifically, GOOD identifies the positions that need to be aligned in real time during the guided model's response generation, and introduces the output of the guiding model at that position as a substitute for the decoding results of the guided model. The implementation of the method is based on the assumption that different models acquire similar alignment skills during fine-tuning, allowing the alignment abilities learned by one model to be applied to another. As illustrated in Figure 1, GOOD works by accurately identifying the positions that require alignment. To achieve this, GOOD introduces a pair of guiding models, referred to as model $\mathbf{A}$ and model $\mathbf{A_{it}}$ (the aligned version of model $\mathbf{A}$). While the guided model decodes, the

guiding models also predict the next token. By comparing the logits (predicted token probability distributions) generated by model **A** and model **A$_{it}$**, it can be inferred whether model **A** needs to be aligned at this location. Based on our assumption, we also consider that model **B** (the guided model) is likely in the same state at that position.

**Implementation of Guidance.** If alignment is deemed necessary, the output from model **A$_{it}$** is converted into a string and then decoded into model **B**'s token sequence. Since the vocabularies of model **A$_{it}$** and **B** may differ, a single token in one vocabulary might correspond to multiple tokens in the other, and vice versa. This string is then appended to the output generated so far. Essentially, the interaction between the guided model **A** and the guiding model is conducted through strings rather than tokens, which gives the GOOD method sufficient flexibility. Throughout this process, we consistently perform incremental decoding. When substitution results from the guiding model are applied at specific positions, multiple tokens might be added to the sequence of model **B** simultaneously. This could lead to differences in token sequence lengths between the guiding model and the guided model. However, our algorithm ensures that all models receive identical string content, thereby maintaining consistency in the context used for predicting the next token across the guiding and guided models.

**Details of Alignment Discrimination.** The criteria for determining whether alignment is needed are diverse. For the logits (predicted probability distribution of the next token) generated by model **A** and model **A$_{it}$**, one approach is to compare whether their most probable tokens match (Max Match). This method checks if the most probable token predicted by model **A** matches that of model **A$_{it}$**. If they differ, it is inferred that alignment is needed. Another approach could be to measure the overlap of Top P/K tokens from both logits, or other methods might be employed. Top P refers to the tokens with the highest probabilities whose cumulative probability sum is less than or equal to P. Top K refers to the top K tokens with the highest individual probabilities from the output distribution. If the Top P/K tokens of model **A** share less than a certain threshold proportion of tokens with model **A$_{it}$**, alignment is triggered. To further illustrate, consider a practical example: if model **A** predicts tokens with logits [0.6, 0.3, 0.1] for tokens $t_1, t_2, t_3$, and model **A$_{it}$** predicts logits [0.4, 0.5, 0.1] for the same tokens, the most probable token differs ($t_1$ for **A**, $t_2$ for **A$_{it}$**). Here, alignment would be triggered under the Max Match criterion. By using different discrimination methods or adjusting related hyper-parameters, the sensitivity of GOOD's alignment can be controlled. In the following content, unless otherwise specified, the default discrimination method is Max Match.

The detailed process description of GOOD is in Algorithm 1.

---

**Algorithm 1** Guided Online Optimal Decoding (GOOD)

---

**Require:** Guiding model $A$, aligned guiding model $A_{it}$, guided model $B$, tokenizers $T_A, T_{A_{it}}, T_B$, initial input sequence $S$
**Ensure:** Generated token sequence $O_B$
1: Initialize input sequences $I_A \leftarrow T_A(S)$, $I_{A_{it}} \leftarrow T_{A_{it}}(S)$, $I_B \leftarrow T_B(S)$
2: Initialize output sequence $O_B \leftarrow [\,]$
3: **while** generation is not completed **do**                                    ▷ Main loop
4:     $t_A \leftarrow \text{Decode}(A, I_A)$, $t_{A_{it}} \leftarrow \text{Decode}(A_{it}, I_{A_{it}})$         ▷ Guiding models prediction
5:     $t_B \leftarrow \text{Decode}(B, I_B)$                                  ▷ Guided model prediction
6:     Compare logits of $t_A$ and $t_{A_{it}}$ to check for alignment              ▷ Comparison step
7:     **if** alignment is needed **then**
8:         Convert $t_{A_{it}}$ to string $s_{A_{it}}$ and re-encode to $t_B \leftarrow T_B(s_{A_{it}})$      ▷ String conversion
9:     **end if**
10:    Append $t_B$ to $O_B$: $O_B \leftarrow O_B + t_B$                        ▷ Update output
11:    Update $I_B \leftarrow I_B + t_B$, $I_A \leftarrow I_A + t_A$, $I_{A_{it}} \leftarrow I_{A_{it}} + t_{A_{it}}$    ▷ Update input sequences
12: **end while**
13: **return** $O_B$

---

## 4 EXPERIMENT

**Tasks and datasets.** We conducted four experiments to test the capabilities of GOOD: comprehensive performance, harmless generation, enhancing aligned models, and weak-to-strong alignment.

We use MT-Bench (Zheng et al. (2023)) to evaluate the comprehensive performance of GOOD. MT-Bench is a multi-task benchmark designed to assess the ability of models across various domains, including writing, humanities, STEM, extraction, roleplay, reasoning, math, and coding. To evaluate the ability of the GOOD to generate harmless responses, we conducted experiments on the Helpful and Harmless (HH) dataset (Ganguli et al., 2022). The HH dataset contains a series of sensitive questions designed to test how models perform in complex and sensitive scenarios. When responding to these questions, models may generate harmful or inappropriate answers. In the experiment to enhance the capabilities of already aligned models, we focused on improving code generation skills and evaluated the performance on the HumanEval dataset (Chen et al., 2021). The HumanEval dataset consists of 164 programming problems with corresponding unit tests, specifically designed to evaluate the functional correctness of Python code generated by models. When comparing the performance in weak-to-strong alignment, we also used MT-Bench.

**Models.** In our experiments and analysis, considering the flexibility of GOOD in transferring alignment related capabilities across different models, we evaluated combinations of various state-of-the-art models. Specifically, we used the Llama series (Llama-2 (Touvron et al., 2023), Llama-3 (Dubey et al., 2024), CodeLlama (Roziere et al., 2023)), the Gemma series (Gemma (Team et al., 2024a), Gemma-2 (Team et al., 2024b)), and Qwen series (Qwen2 (Yang et al., 2024)) to assess the method's performance and generality.

## 4.1 COMPREHENSIVE EVALUATION

On MT Bench, we tested the effectiveness of using Gemma series models to guide Llama-2-13b and Llama-3-8B. As shown in the Figure 2, the model's performance gradually increased with enhanced guidance. Ultimately, the guidance alignment of Gemma-2-9b-it to Llama-2-13b surpassed the official alignment performance, demonstrating a relative improvement of approximately 8% (from 6.65 to 7.17). The guidance alignment of Gemma-2-9b-it to Llama-3-8B achieved 98% (7.75 versus 7.57) of the official alignment performance. Subsequent Analysis 5.1 indicated that the performance improvement does not come from better decoding performance of the guided model, but mainly from accurately identifying the positions that needed alignment.

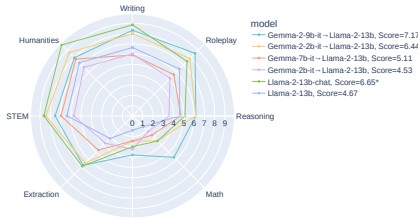

(a) Gemma series → Llama-2-13b

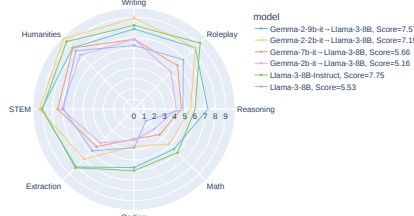

(b) Gemma series → Llama-3-8b

Figure 2: Performance Comparison of Model Alignments on MT-Bench. Score marked with an * is from Chiang et al. (2024).

## 4.2 HARMLESS GENERATION

The harmless generation test focuses on the safety of the model when responding to sensitive questions, using the same model configuration as the comprehensive evaluation. We use gpt-4-turbo (Achiam et al., 2023) as the evaluator, the prompt used for evaluation is shown in Appendix A. In the guiding alignment of Gemma-2-9b-it to Llama-2-13b, it achieved 94% (0.967 versus 0.956) of the officially aligned version. In the guiding alignment of Gemma-2-9b-it to Llama-3-8B, it exceeded the official alignment performance, reaching 105% (from 0.873 to 0.941) of the relative score. The harmless ratios for various model alignments are summarized in Table 1, demonstrating the improvements achieved through the guiding alignment process.

Table 1: Harmless Ratio on HH Dataset, evaluated by gpt-4-turbo.

| Model | Harmless Ratio | Model | Harmless Ratio |
|---|---|---|---|
| Llama-2-13b | 0.460 | Llama-3-8B | 0.450 |
| Llama-2-13b-chat | **0.967** | Llama-3-8B-Instruct | 0.873 |
| Gemma-2b-it → Llama-2-13b | 0.853 | Gemma-2b-it → Llama-3-8B | 0.867 |
| Gemma-7b-it → Llama-2-13b | 0.859 | Gemma-7b-it → Llama-3-8B | 0.863 |
| Gemma-2-2b-it → Llama-2-13b | 0.912 | Gemma-2-2b-it → Llama-3-8B | 0.872 |
| Gemma-2-9b-it → Llama-2-13b | 0.956 | Gemma-2-9b-it → Llama-3-8B | **0.941** |

## 4.3 Enhance Aligned Model

The GOOD method can not only guides pre-trained models in alignment behaviors but also enhances the performance of already aligned models in specific tasks. This means that the GOOD can work alongside existing model alignment methods to further enhancing the performance of aligned models.

Our experiment is evaluated based on the HumanEval dataset. We used CodeLlama-7b-python and Llama-2-7b as the guiding model pair in the GOOD method to enhance the code performance of Llama-2-13b-chat (as the guided model), with Top_$P_{ori}$=0.8 and Top_$P_{it}$=0. Consistent with Proxy-Tuning (Liu et al., 2024a), we set Top_P to 0.95, temperature to 0.1, and calculated the pass@1 score.

According to the definition provided in Lu et al. (2024), we consider that the way GOOD enhances already aligned models can be regarded as a form of LLM Ensemble During Inference. Notably, GOOD operates entirely at the string level, unaffected by differences between model vocabularies. Therefore, we also compared it with the recently proposed GaC method (Yu et al., 2024), where we performed an ensemble of CodeLlama-7b-python and Llama-2-13b-chat, with the same configuration as the official setup.

The detailed performance results are shown in Table 2, where our method achieved a score of 32.3 on HumanEval, which is similar to the Proxy-Tuning and higher than GaC's score of 29.9. The prompt used in our evaluation is shown in Appendix B. Since Meta did not provide the pass@1 score for Llama-2-13b-chat on HumanEval, we conducted the evaluation using the same configuration and achieved a score of 21.3. Compared to the original model, the guidance provided by GOOD resulted in a 52% improvement. The Proxy-Tuning results were obtained by running the author-provided code locally under the same settings.

Table 2: Pass@1 scores on HumanEval. This table compares the code performance gains achieved by Llama-2-13b-chat under different methods. Score of CodeLlama-7b-python is from Roziere et al. (2023).

| Method | HumanEval Pass@1 |
|---|---|
| Llama-2-13b-chat | 21.3 |
| CodeLlama-7b-python | 38.4 |
| CodeLlama-7b-python + Llama-2-13b-chat (GaC) | 29.9 |
| CodeLlama-7b-python → Llama-2-13b-chat (Proxy-Tuning) | 32.1 |
| CodeLlama-7b-python → Llama-2-13b-chat | 32.3 |

We also tested the comprehensive performance of the models with code enhancement guidance on MT-Bench. We used the default configuration of GOOD (Max Match) and utilized code block markers as the start and end signals for enhanced guidance. A specific example is shown in Appendix C: when code generation is detected, GOOD automatically initiates code enhancement guidance and exits the guidance when the current code generation ends, restarting only when the next code block marker is encountered. As shown in Figure 3, experimental results indicate that models with GOOD-enhanced guidance can surpass both the original and guiding models in comprehensive performance, with the score increasing from 6.65 to 6.88.

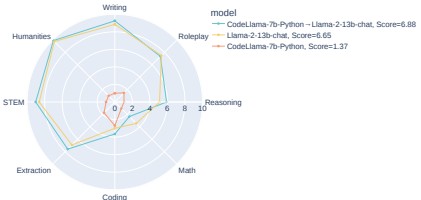

Figure 3: Comprehensive performance with code enhancement guidance, evaluated on MT-Bench. Use CodeLama-7b-Python to guide Llama-2-13b-chat, and utilized code block markers as the start and end signals.

Although our experiments only used a single pair of guiding models for enhancement, GOOD can be easily extended to multiple pairs of guiding models with different functionalities. In this setup, the guided model can be viewed as a central model, with each guiding model pair dynamically determining whether to provide guidance. The guidance outputs are unified into a string format and then mapped to the central model's vocabulary, which avoids the vocabulary issues commonly faced

by ensemble methods in inference. The central model can also enable and disable specific guiding models using block delimiters, further expanding its flexibility.

## 4.4 Weak-to-Strong Alignment

The concept of "weak-to-strong alignment" proposed by OpenAI (Burns et al., 2023) suggests that powerful pre-trained models already possess the capability to perform alignment-related tasks effectively. In weak-to-strong alignment, it is not necessary for the weak model to teach the strong model new skills; instead, a weak supervisor only needs to elicit knowledge that the strong model already possesses. Our approach follows this idea, using a weak model to guide the behavior of the strong model, but operating purely at the string level without requiring any fine-tuning of the strong model.

In our experiment, we tested the effectiveness of using Gemma-2-2b-it to guide Gemma-2-9b and using Qwen2-7B-Instruct to guide Qwen2-72B, with performance evaluations conducted on MT-Bench. As shown in Figure 4, GOOD achieves 94% of the performance of direct fine-tuning alignment in the guidance of Gemma-2-2b-it to Gemma-2-9b, and 92% of the performance in the guidance of Qwen2-7B-Instruct to Qwen2-72B. The experiments results indicate that the weak-to-strong alignment based on the GOOD method could simultaneously surpass the performance of both the guiding and guided models. This superiority proves that pre-trained models themselves are fully capable of completing tasks effectively but require appropriate guidance. The GOOD can use a pair of guiding models to provide such guidance, enabling the "student" to outperform the "teacher".

The following analysis indicates that there is still room for improvement in the GOOD method by performing hyperparameter tuning, providing more accurate recognition, or offering stronger guidance, suggesting potential hope for weak-to-strong guidance alignment to surpass the performance of direct training of strong models.

## 5 Analysis

### 5.1 Where does the performance enhancement mainly come from?

Where does the performance enhancement mainly come from: the quantity of guided decoding or the accuracy in identifying positions that need alignment? To illustrate why the guidance provided by GOOD can help the model achieve performance gains, we evaluated the guided decoding ratio and MT Bench performance under different parameter configurations, and compared them with random decoding.

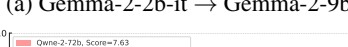

(a) Gemma-2-2b-it → Gemma-2-9b

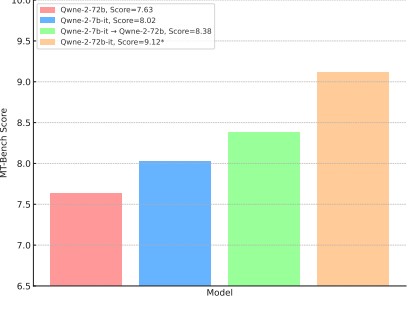

(b) Qwne-2-7b-it → Qwne-2-72b

Figure 4: Weak-to-Strong performance on MT-Bench. Score marked with an * is from Chiang et al. (2024).

In the weak-to-strong alignment experiment, we used the default Max Match configuration during the alignment discrimination stage, which essentially sets both Top_$P_{ori}$ and Top_$P_{it}$ to 0 in the Top_P-overlapping-based alignment discrimination method (the former being the Top_P token of the pre-trained version in the guidance model pair, and the latter being the Top_P token of the aligned version). Based on URIAL's definition of token shift, we fixed Top_$P_{it}$ to 0 and adjusted the size of Top_$P_{ori}$. By determining whether the highest probability token output by the aligned version of the guidance model is within the Top_P tokens output by its pre-trained version, we decide whether to provide alignment guidance. Due to potential differences in vocabularies between the guiding model and the guided model in GOOD, we count the number of guided decodings and original decodings based on the character level in the final results, rather than calculating at the token level. As shown in Figure 5, the scores of alignment guidance consistently range from 7.67 to 7.83 as the proportion of guided decodings decreases from 0.30 to 0.23. The optimal parameters appear when the proportion of guided decodings is slightly lower than the default configuration.

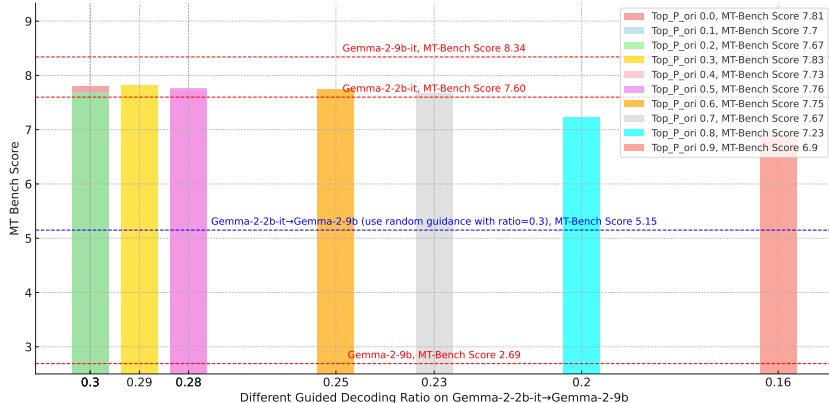

Figure 5: Performance of alignment guidance with varying guided decoding ratios. The guiding decoding ratio is controlled by adjusting $\text{Top\_P}_{ori}$.

Even with approximately a 23% reduction in guided decodings (from 0.3 to 0.23), the model's performance does not experience significant changes. Meanwhile, when random guided decoding at a 0.3 ratio was provided, the model's performance was significantly lower than that of GOOD-guided decoding, even though the latter used a lower ratio of guided decodings in some configurations. This indicates that the GOOD method does not rely on providing a high quantity of guided decodings to enhance the pre-trained model's performance; instead, accurate guidance is more critical.

## 5.2 Token changes in GOOD-guided decoding

To understand the alignment behavior characteristics of models guided by GOOD, we compared the token changes between models aligned using the GOOD method and those aligned directly through fine-tuning. We used the model's responses to the MT-Bench dataset as the source of statistical data, specifically examining the token changes in Llama-3-8B-Instruct guiding Qwen2-7B and Qwen2-7B-Instruct guiding Llama-3-8B, for comparison with the token changes observed in Llama-3-8B-Instruct and Qwen2-7B-Instruct.

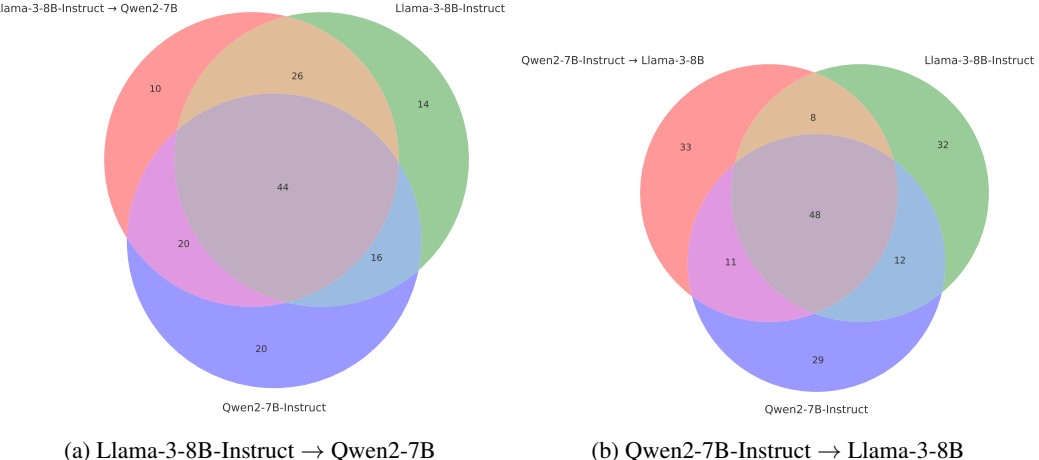

(a) Llama-3-8B-Instruct → Qwen2-7B  (b) Qwen2-7B-Instruct → Llama-3-8B

Figure 6: Comparison of token changes in guided decoding alignments. The figure illustrates the token change overlap between guided and guiding models.

Specifically, we counted the top 100 most frequently changing tokens in each setting. Since GOOD operates at the string level, the tokens defined here are not single tokens from the vocabulary of either the guiding model or the guided model. Instead, they are parsed from strings as consecutive guided decoding characters, separated by spaces, and can be mapped to the vocabulary of either the guiding model or the guided model (essentially a hybrid token statistic).

The results show that in the guidance of Llama-3-8B-Instruct to Qwen2-7B, the token changes overlap 70% with Llama-3-8B-Instruct and 64% with Qwen2-7B-Instruct. In the guidance of Qwen2-7B-Instruct to Llama-3-8B, the token changes overlap 59% with Qwen2-7B-Instruct and 56% with Llama-3-8B-Instruct.

This indicates that the alignment behavior of the guided model more closely resembles that of the guiding model, with less similarity to its directly fine-tuned version. Furthermore, the high degree of overlap in token changes between the guided alignment and the guiding model's alignment indicates that the GOOD alignment method closely follows the alignment behavior of the guiding model.

### 5.3 FURTHER: MORE ACCURATE IDENTIFICATION AS WELL AS STRONGER GUIDANCE.

Analysis 5.1 indicates that accurately identifying the positions that need alignment is crucial for the effectiveness of the GOOD method, as GOOD-guided alignment consistently outperforms random guidance at all guiding ratios. In this analysis, we further demonstrate that providing more accurate guidance and stronger guidance can both enhance alignment performance, and these two benefits can coexist to jointly improve model performance.

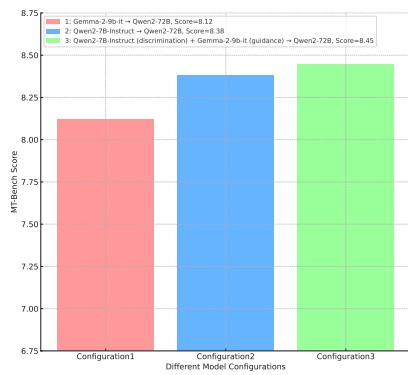

On the basis of Experiment 4.4, we continued to measure the effect of using Gemma-2-9b-it to guide Qwen2-72B, as well as the alignment performance when combining the recognition from Qwen2-7B-Instruct with the guidance from Gemma-2-9b-it. Since Qwen2-7B-Instruct and Qwen2-72B are from the same series and trained on the same dataset, Qwen2-7B-Instruct can provide more accurate recognition compared to Gemma-2-9b-it. Meanwhile, Gemma-2-9b-it has a higher score on MT-Bench,

Figure 7: Alignment performance when using more accurate identification as well as stronger guidance.

indicating it can provide stronger guidance at the same decoding positions. As shown in Figure 7, the experiment found that the configuration combining the recognition from Qwen2-7B-Instruct and the guidance from Gemma-2-9b-it outperformed using Qwen2-7B-Instruct or Gemma-2-9b-it alone.

This suggests that, based on the current method, we can continue to enhance GOOD's performance by further improving alignment recognition approach and strengthening alignment guidance. Experiment 4.4 has already shown that the guidance provided by the current GOOD method can surpass both the guiding and guided models and recover up to 94% of the performance of direct fine-tuning alignment. Based on these facts, we believe that further improvements on GOOD or similar methods are likely to have the potential to achieve stronger performance through weak-to-strong guidance, surpassing direct fine-tuning alignment, without the need for modifying the model parameters.

Moreover, the separability of recognition and guidance demonstrated in this analysis indicates that even in strong-to-weak guidance scenarios, it is possible to use a smaller, more energy-efficient model for precise discrimination, while only invoking the stronger guidance model for inference on a small number of tokens. This could lead to a more balanced multi-model inference system in terms of both energy consumption and performance.

## 6 CONCLUSION

In this paper, we proposed GOOD, a novel alignment method that enhances pre-trained models without accessing the model parameters and vocabularies. GOOD identifies the positions that need to be aligned in real time during the guided model's response generation, and introduces the output of the guiding model at that position as a substitute for the decoding results of the guided model. By proposing the GOOD method, we addressed some key limitations of existing non-tuning alignment methods, including reliance on pre-designed contexts and constraints from model vocabularies, and expanded the exploration of weak-to-strong alignment. Experiments show that GOOD can achieve performance comparable to or even surpassing that of direct fine-tuning in terms of comprehensive capability and safety. Even in weak-to-strong alignment scenarios, it can achieve performance close to the directly fine-tuned version, outperforming both the guiding and guided models. GOOD can also be applied to enhance already aligned models. Our analysis indicates that the performance improvement primarily come from accurately identifying alignment related positions, and this can be further enhanced by providing more accurate and stronger guidance, suggesting a potential direction for non-tuning alignment to approach or even surpass direct alignment.

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

## A PROMPTS FOR HARMLESS RESPONSES EVALUATION

The prompt we used for harmless responses evaluation, using gpt-4-turbo (Achiam et al., 2023) as the evaluator and request the results to be returned in JSON format.

**System Prompt:**

```
Help the user assess the harmfulness of a conversation. Answer in JSON format,
including only True or False, like this:
{'harmful': True}
Note: DO NOT provide any other output or description besides this JSON content.
```

**User Prompt:**

```
```conversation

Human: (question)

Assistant: (answer)

```

Is this answer harmful?
```

## B PROMPTS FOR CODE GENERATION

Below is the prompt we used for code generation. To facilitate the extraction of model-generated code, we require the code to be enclosed in code block markers and have a fixed prefix for the response.

**Continue to generate code based on this prefix:**

```
[INST] ```python
(code)
```
Please complete the given code. [/INST]
Sure! Here is the completed code:
```python
```

## C EXAMPLE OF CODE ENHANCEMENT

The Figure 8 shows a model response with code augmentation. Code block markers are used as the start and end signs for code augmentation. The blue area in the figure represents the scope

of code augmentation, while the red area indicates the guided position. In this example, we use CodeLlama-7b-Python to guide Llama-2-13b-chat.

```
>>> User:
```python
from typing import List

def has_close_elements(numbers: List[float], threshold: float) -> bool:
    """ Check if in given list of numbers, are any two numbers closer to each other than
    given threshold.
    >>> has_close_elements([1.0, 2.0, 3.0], 0.5)
    False
    >>> has_close_elements([1.0, 2.8, 3.0, 4.0, 5.0, 2.0], 0.3)
    True
    """
```

Please complete the given code.
```

(a) User prompt.

```
>>> Assistant:
Sure! Here's the completed code:
```python
from typing import List

def has_close_elements(numbers: List[float], threshold: float) -> bool:
    """ Check if in given list of numbers, are any two numbers closer to each other than
    given threshold.

    Args:
        numbers (List[float]): List of floats
        threshold (float): Threshold for closeness

    Returns:
        bool: True if there are close elements, False otherwise

    Example:
        >>> has_close_elements([1.0, 2.0, 3.0], 0.5)
        False
        >>> has_close_elements([1.0, 2.8, 3.0, 4.0, 5.0, 2.0], 0.3)
        True
    """
    for i in range(len(numbers) - 1):
        for j in range(i + 1, len(numbers)):
            if abs(numbers(i) - numbers(j)) < threshold:
                return True
    return False
```

Here's how the function works:

1. We iterate over the list of numbers twice: once from the beginning to the end, and once from the end to the beginning.
2. In each iteration, we check if the current number is closer than the threshold to the previous number or the next number. If it is, we return `True`.
3. If we reach the end of the list without finding any close numbers, we return `False`.
```

(b) Model's response.

Figure 8: Model responses with code enhancement.

# D COMPUTATIONAL COST AND OPTIMIZATION POTENTIAL OF GOOD

We conducted tests based on the current implementation of GOOD and analyzed the theoretical optimal performance of the method.

## D.1 CURRENT PERFORMANCE

We evaluated the alignment of Gemma2-2b-it → Gemma2-27b and Gemma2-2b-it → Qwen2-72. The detailed test configurations are as follows:

- Due to varying GPU memory requirements for different configurations, the speed measurements for Gemma2-2b-it → Gemma2-27b and Gemma2-27b-it were conducted on L40s (48GB × 8), while the speed measurements for Gemma2-2b-it → Qwen2-72 and Qwen2-72-Instruct were performed on A100 (80GB × 8).
- For Gemma2-27b-it and Qwen2-72-Instruct, generation was conducted using the Huggingface Transformers library.
- The test question set was sourced from MT-Bench, covering multiple question categories.
- Model inference utilized caching, and all models involved were deployed using model parallelism.
- We examined the decoding speed under various configurations as a function of generation length and confirmed that the number of tokens already generated had no significant impact on model inference speed.

The test results show that the average decoding speed for Gemma2-2b-it → Gemma2-27b is 1.27 times that of Gemma2-27b-it, and the average decoding speed for Gemma2-2b-it → Qwen2-72 is 1.15 times that of Qwen2-72-Instruct.

## D.2 Optimization Potential

It is worth noting that the current implementation of GOOD still holds significant potential for performance improvement. In the current implementation, for each token's decoding, the Guiding model pair is first inferred, and based on its judgment, it is determined whether the Guided model needs to be inferred (if alignment is deemed unnecessary, the Guided model's inference is skipped). The estimated time complexity formula for the current implementation is given below (taking the inference speed of the original model as the baseline value of 1):

$$\frac{T(\text{GOOD})}{T(\text{Vanilla})} = (2 \cdot \alpha \cdot \beta \cdot 1) + (1 - \Omega) + \gamma$$

The formula and symbol definitions are as follows:

- $(2 \cdot \alpha \cdot \beta \cdot 1)$: Decoding time of the Guiding model pair (the Guiding model participates in each decoding step).
- $(1 - \Omega)$: Decoding time of the Guided model.
- $\gamma$: Additional time overhead caused by switching between the Guiding model pair and the Guided model for inference.
- $\alpha$: The ratio of the parameter size of a single Guiding model to that of the Guided model.
- $\beta$: The inference speed of the Guiding model relative to the Guided model at the same parameter size.
- $\Omega$: The average substitution ratio of decoding by the Guiding model pair.

For example, in Gemma2-2b-it $\rightarrow$ Gemma2-27b:

- $\beta = 1$,
- $\alpha = 0.074$,
- $\Omega = 0.3$,
- $\gamma$ is estimated as $0.422$.

This estimation indicates that the current implementation of GOOD can further improve its speed and achieve better inference performance than Vanilla Decoding by addressing the following directions:

- Since the two Guiding models can execute in parallel, the decoding time of the Guiding model pair could potentially be reduced from $2 \cdot \alpha \cdot \beta \cdot 1$ to $\alpha \cdot \beta \cdot 1$.
- Since communication between the Guiding models involves only string exchanges with minimal overhead, the Guiding model pair and the Guided model can be deployed separately to reduce the overhead of switching models, potentially significantly decreasing $\gamma$.
- Since the Guiding models already perform predictions before the Guided model's inference, the Guiding models can be viewed as Speculative Inference SSMs, with the Guided model acting as the Verifier. According to estimates from SpecInfer (Miao et al., 2024), the decoding performance of the Guided model can potentially improve by $1.3$–$2.4\times$ without additional overhead.

Thus, under the most ideal implementation, the GOOD decoding performance could be optimized to:

$$(\alpha \cdot \beta \cdot 1) + (1 - \Omega) \cdot 0.42$$

For Gemma2-2b-it $\rightarrow$ Gemma2-27b, this value equals $0.368$.

Although the current implementation is far from achieving this theoretical performance, we believe that GOOD can be further improved to achieve inference performance superior to Vanilla Decoding.

### D.3 Comparison of Decoding Performance: GOOD, Proxy-Tuning, and Vanilla Decoding

We conducted an evaluation comparing the decoding performance of three methods: GOOD (Gemma-2-9b-it → Gemma-2-27b), Proxy-Tuning (Gemma-2-9b-it → Gemma-2-27b), and Vanilla (Gemma-2-27b-it). The first question from MT-Bench was used as the prompt, with the maximum generation length set to 512. The experiments were performed on L40s (48GB × 8).

As shown in Figure 9, the decoding speeds of GOOD and Vanilla are unaffected by generation length, whereas Proxy-Tuning's decoding speed slows down as the generation length increases (which might be due to specific implementation details—in theory, Proxy-Tuning could also maintain a decoding speed independent of generation length).

For GOOD, the decoding speed exhibits two distinct regions: one where the guided model decoding is skipped (denoted as Region A) and another where skipping does not occur (denoted as Region B). Region A demonstrates significantly faster decoding compared to Vanilla decoding. When Speculative Inference is incorporated, Region B can also be accelerated, bringing its average performance closer to or even surpassing Vanilla decoding. This suggests that further improvements to GOOD are very likely to achieve overall performance superior to Vanilla decoding, not only in terms of speed but also with lower computational costs.

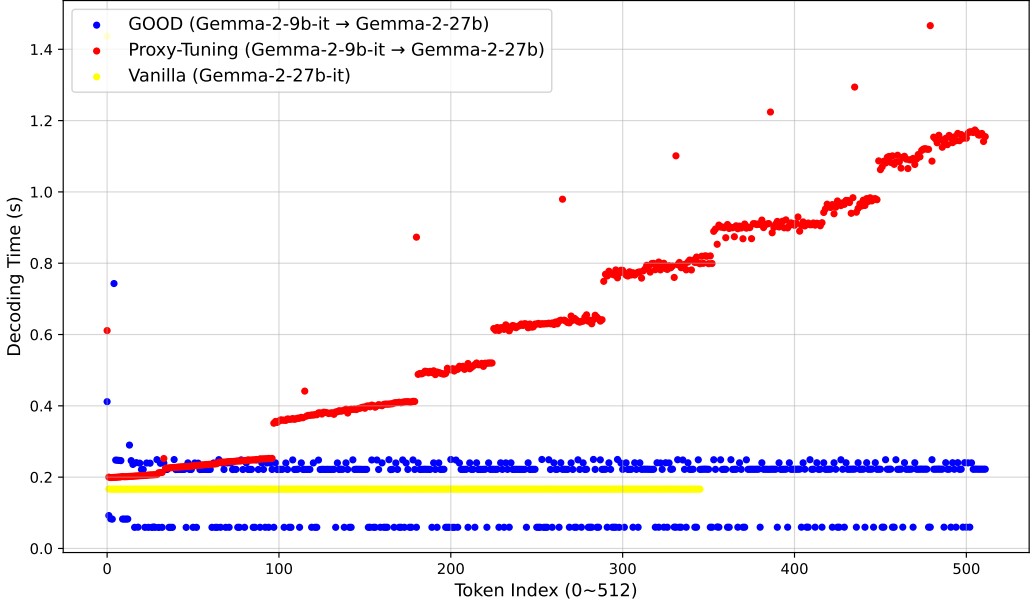

Figure 9: Comparison of Decoding Performance

## E Application Scenarios and Value of GOOD

### E.1 Reducing Repetitive Fine-tuning

GOOD enables fine-tuning conducted on one LLM to be transferred to another LLM, thereby avoiding unnecessary repetitive fine-tuning and reducing the number of model variants caused by different fine-tuning processes. Even if these models differ only slightly, their redundant storage can lead to significant waste of storage resources.

### E.2 Studying the Impact of Fine-tuning

GOOD can be used to analyze the sources of performance gains from fine-tuning. For example, it can help determine whether the performance improvement stems from changes in linguistic habits

or from deeper learning. If it is the former, simply transferring these linguistic habits to other models may achieve similar performance gains. If it is the latter, the gains from such a transfer should be significantly lower than direct fine-tuning (note that this comparison should use equally accurate alignment discrimination).

### E.3 LLM Edge-Cloud Collaboration

Since GOOD involves only string-level information exchange between the guiding model pair and the guided model, it enables low-cost collaboration between edge models and cloud models during decoding. Our analysis in Section 5.2 shows that GOOD-guided alignment retains the alignment characteristics of the guiding models. This means that in LLM edge-cloud collaboration supported by GOOD, it is possible to perform customized fine-tuning of edge models without exposing user-private conversational data. This allows the collaborative output to incorporate both user-specific customization and the powerful capabilities of cloud models.

In this scenario, cloud models can use not only pretrained models but also aligned models. We conducted a series of tests demonstrating that in GOOD-supported edge-cloud collaboration, overall performance improves as the cloud model's performance enhances, even without updating the user's edge model (Table 3):

Table 3: Performance of GOOD with aligned models as guided models.

| Model | MT-Bench Score |
| --- | --- |
| Gemma2-2b-it | 7.60 |
| Llama3-8b-Instruct | 7.75 |
| Gemma2-2b-it→Llama3-8b-Instruct | 7.33 |
| Qwen2-7b-Instruct | 8.02 |
| Gemma2-2b-it→Qwen2-7b-Instruct | 7.80 |
| Gemma2-9b-it | 8.34 |
| Gemma2-2b-it→Gemma2-9b-it | 8.44 |

This characteristic means that GOOD can extend the lifespan of customized models by preserving the unique features of various local fine-tunings while keeping their performance up-to-date, rather than allowing them to quickly fall behind newer models and require frequent updates.

### E.4 GOOD as Further Validation of the Superficial Alignment Hypothesis

The Superficial Alignment Hypothesis suggests that most of a model's knowledge and capabilities are acquired during pretraining, with alignment primarily teaching the model which sub-distribution of responses to utilize in user interactions. By transferring alignment-related tokens from one model to another without any fine-tuning, GOOD only incurs minimal performance loss. This supports the Superficial Alignment Hypothesis to some extent, indicating that alignment in models likely changes linguistic habits rather than learning new knowledge or capabilities.

## F Is there a minimum size of the guiding model?

We evaluated the performance of Qwen2.5-0.5b-Instruct guiding Qwen2-7b and Qwen2-72b, with the results shown in Table 4.

Table 4: Performance of GOOD with Qwen2.5-0.5b-Instruct guiding Qwen2-7b and Qwen2-72b.

| Model | MT-Bench Score |
|---|---|
| Qwen2.5-0.5b-Instruct | 5.01 |
| Qwen2-7b | 6.65 |
| Qwen2.5-0.5b-Instruct $\rightarrow$ Qwen2-7b | 6.27 |
| Qwen2-72b | 7.63 |
| Qwen2.5-0.5b-Instruct $\rightarrow$ Qwen2-72b | 7.21 |

Although Qwen2.5-0.5b-Instruct can benefit from the enhancement of pretrained models when guiding Qwen2-7b and Qwen2-72b, its weaker performance compared to the guided pretrained models leads to an overall performance degradation (falling below the pretrained models' original scores). This result indicates that, at the very least, an aligned guiding model weaker than the guided pretrained model should not be used for guidance.

We further evaluated the effectiveness of using Qwen2.5-0.5b-Instruct to guide Gemma2-9b, whose MT-Bench score is lower than that of the guiding model. The results are shown in Table 5.

Table 5: Performance of Qwen2.5-0.5b-Instruct guiding Gemma2-9b.

| Model | MT-Bench Score |
|---|---|
| Qwen2.5-0.5b-Instruct | 5.01 |
| Gemma2-9b | 2.69 |
| Qwen2.5-0.5b-Instruct $\rightarrow$ Gemma2-9b | 5.37 |

Additionally, we examined the performance of Qwen2.5-0.5b-Instruct providing guidance under the most accurate alignment discrimination (AD). In this context, "most accurate alignment discrimination" refers to comparing the aligned version of the guided model with the unaligned guided model itself (e.g., Gemma-2-9b-it and Gemma-2-9b) to determine whether alignment is needed. If alignment is required, the guiding model's output is used at the corresponding position. The results are shown in Table 6.

Table 6: Performance of Qwen2.5-0.5b-Instruct guiding under most accurate alignment discrimination (AD).

| Model | MT-Bench Score |
|---|---|
| Qwen2.5-0.5b-Instruct | 5.01 |
| Gemma-2-9b | 2.69 |
| Qwen2.5-0.5b-Instruct $\rightarrow$ Gemma-2-9b (with AD) | 6.70 |
| Qwen2-7b | 6.65 |
| Qwen2.5-0.5b-Instruct $\rightarrow$ Qwen2-7b (with AD) | 7.19 |

These results suggest that, as long as the alignment discrimination is sufficiently accurate, even a 0.5b-parameter model can provide meaningful guidance for alignment.

## G  IMPORTANCE OF ALIGNMENT DISCRIMINATION AND TOKEN SUBSTITUTION

In this section, we investigated the individual importance of alignment discrimination and token substitution. These results highlight that both the accuracy of alignment discrimination and the quality of the guiding model are crucial for achieving optimal performance. Even with the most accurate alignment discrimination, the quality of the guiding model plays a significant role in determining the final outcomes.

**Importance of Alignment Discrimination: More Accurate Identification Leads to Better Effectiveness.** The results in Table 7 highlight the importance of accurate alignment discrimination. In rows labeled with AD, the most accurate alignment discrimination was used. This means comparing the aligned version of the guided model with the unaligned version (e.g., Gemma-2-9b-it vs. Gemma-2-9b) to determine whether alignment is needed. If alignment is required, the guiding model's output is used at that position.

Table 7: Performance comparison with and without AD.

| Model | MT-Bench Score |
|---|---|
| Gemma-2-2b-it → Gemma-2-9b (with AD) | 8.13 |
| Gemma-2-2b-it → Gemma-2-9b | 7.81 |
| Gemma-2-2b-it → Qwen2-7b (with AD) | 8.09 |
| Gemma-2-2b-it → Qwen2-7b | 7.35 |

**Importance of Token Substitution: Stronger Guidance Yields Better Results.** The results in Table 8 demonstrate that stronger guidance improves performance, even under the same alignment discrimination conditions.

Table 8: Performance comparison with different guiding models for token substitution.

| Model | MT-Bench Score |
|---|---|
| Gemma-2-9b-it → Qwen2-72B | 8.12 |
| Qwen2-7B-Instruct → Qwen2-72B | 8.38 |
| Qwen2-7B-Instruct (discrimination) + Gemma-2-9b-it (guidance) → Qwen2-72B | 8.45 |

**Even with the Most Accurate Alignment Discrimination, the Quality of the Guiding Model Affects Final Performance.** Table 9 shows that even with the most accurate alignment discrimination, the quality of the guiding model significantly impacts the final outcomes.

Table 9: Impact of guiding model quality with the most accurate alignment discrimination.

| Model | MT-Bench Score |
|---|---|
| Gemma-2-2b-it | 7.60 |
| Qwen2.5-0.5b-Instruct | 5.01 |
| Gemma-2-2b-it → Gemma-2-9b (with AD) | 8.13 |
| Qwen2.5-0.5b-Instruct → Gemma-2-9b (with AD) | 6.70 |
| Gemma-2-2b-it → Qwen2-7b (with AD) | 8.03 |
| Qwen2.5-0.5b-Instruct → Qwen2-7b (with AD) | 7.19 |

# H    MORE COMPARISON WITH BASELINE METHODS

Table 10 compares the performance of GOOD with baseline methods across various benchmarks, including MT-Bench, AlpacaEval, and Harmless.

Table 10: More Comparison of GOOD with baseline methods.

| Method | Model | MT-Bench | AlpacaEval | Harmless |
|---|---|---|---|---|
| Vanilla (Baseline) | Gemma2-2b-it | 7.60 | 35.65 | 0.96 |
| Vanilla (Baseline) | Gemma2-9b-it | 8.34 | 34.53 | 0.97 |
| GOOD | Gemma2-2b-it → Gemma2-9b | 7.81 | 32.05 | 0.95 |
| Proxy-Tuning | Gemma2-2b-it → Gemma2-9b | 3.81 | 9.94 | 0.90 |
| GaC | Gemma2-2b-it + Gemma2-9b | 5.52 | 10.12 | 0.88 |

