# OpenReview forum: "GOOD: Decoding-Time Black-Box LLM Alignment"
_ICLR.cc/2025/Conference — ICLR 2025 Conference Withdrawn Submission_

### Official Review · Reviewer_KAGp · 2024-10-16

**Soundness:** 2
**Presentation:** 3
**Contribution:** 2
**Rating:** 5
**Confidence:** 4

**Summary:**

This paper proposes Guided Online Optimal Decoding (GOOD), a training-free and flexible decoding framework for aligning large language models (LLMs). This approach is based on the assumption that different models develop similar alignment skills during fine-tuning.

During each inference step, (1) GOOD first generates a token using a guiding model pair, which consists of a base pre-trained model and a well-tuned instruct version. (2) It then decodes the token using the guiding model. (3) If alignment is necessary, it converts the token to a string, and re-encodes it using the guiding model's tokenizer. (4) Finally, it updates the output and input sequences.

Extensive experiments on MT-bench, harmless generation, coding and weak-to-strong scenarios and further analysis demonstrate that the effectiveness of GOOD.

**Strengths:**

The paper is well-written and easy to read. The proposed method achieves competitive performance under the experimental setup of this paper, demonstrating results comparable to the instruct version without requiring any training.

**Weaknesses:**

1. The main weakness of the proposed method in this paper is the significantly high inference overhead. In the process of generating each token, three models (the guiding model pair and the guided model) perform synchronous inference, and after each reasoning step, a determination is made to check if alignment is needed. If alignment is required, the results must be concatenated and updated. Therefore, I believe the method proposed in this paper is somewhat tricky, as it increases the number of models involved in inference for the sake of a training-free gimmick, leading to significantly higher inference costs that hinder practical application.

2. The paper lacks an analysis of the inference costs, particularly in comparison to single-model inference and similar methods (like Proxy-Tuning) regarding memory usage and inference time.

3. The paper lacks experimental comparisons. While the authors compare GOOD and Proxy-Tuning in code generation, it is unclear why no comparisons were made for the other two experimental categories (MT-bench and harmless generation). Although Proxy-Tuning has limited flexibility, I believe it is important to conduct a comprehensive comparison, as it serves as a strong baseline for the method proposed in this paper.

4. The paper lacks additional validation, such as further testing of conversational and instruction-following abilities on AlpacaEval, as well as more extensive verification of complex reasoning in mathematics.

5. The font size of the legends and coordinates in the figures is too small, making it difficult for readers to clearly see the information.

6. I believe that the tables in the paper should primarily use a three-line format, as the current table style does not seem to align with academic conventions.

**Questions:**

I would be very happy to engage in further discussion with the authors. My main questions are as follows:

1. I hope to discuss the value of this paper with the authors. In the first point under Weaknesses, I mentioned that the simultaneous inference of three models in the GOOD method may lead to excessively high inference costs, making practical application difficult. Therefore, I would like to gain deeper insights into how this work can inspire future improvements.

2. Could you provide a detailed comparison of inference costs to give readers a comprehensive and intuitive understanding?

3. Could you include comparisons with existing baselines in the experiments of this paper?

4. Could you add more evaluations, such as AlpacaEval or mathematical reasoning?

I will review the paper from a critical perspective to help make it more solid and I look forward to further discussions with you.

---

> ### Author Response · Authors · 2024-11-24
> **Response to Reviewer KAGp: W1, W2, Q1, Q2**
>
> We sincerely appreciate Reviewer KAGp's time, effort, insightful feedback, and constructive suggestions, which are valuable in helping us refine our work. Below, we address the concerns raised and provide additional discussions and justifications to clarify our contribution.
>
> ## W1, W2, Q1, Q2：
>
> We conducted tests based on the current implementation of GOOD and analyzed the theoretical optimal performance of the method.
>
> ### Current Performance
>
> We evaluated the alignment of Gemma2-2b-it → Gemma2-27b and Gemma2-2b-it → Qwen2-72. The detailed test configurations are as follows:
>
> - Due to varying GPU memory requirements for different configurations, the speed measurements for Gemma2-2b-it → Gemma2-27b and Gemma2-27b-it were conducted on L40s (48GB × 8), while the speed measurements for Gemma2-2b-it → Qwen2-72 and Qwen2-72-Instruct were performed on A100 (80GB × 8).
> - For Gemma2-27b-it and Qwen2-72-Instruct, generation was conducted using the Huggingface Transformers library.
> - The test question set was sourced from MT-Bench, covering multiple question categories.
> - Model inference utilized caching, and all models involved were deployed using model parallelism.
> - We examined the decoding speed under various configurations as a function of generation length and confirmed that the number of tokens already generated had no significant impact on model inference speed.
>
> The test results show that the average decoding speed for Gemma2-2b-it → Gemma2-27b is 1.27 times that of Gemma2-27b-it, and the average decoding speed for Gemma2-2b-it → Qwen2-72 is 1.15 times that of Qwen2-72-Instruct.
>
> ### Optimization Potential
>
> It is worth noting that the current implementation of GOOD still holds significant potential for performance improvement. In the current implementation, for each token's decoding, the Guiding model pair is first inferred, and based on its judgment, it is determined whether the Guided model needs to be inferred (if alignment is deemed unnecessary, the Guided model's inference is skipped). The estimated time complexity formula for the current implementation is given below (taking the inference speed of the original model as the baseline value of 1):
>
> T(GOOD)/T(Vanilla) = (2 * α * β * 1) + (1 - Ω) + γ
>
> The formula and symbol definitions are as follows:
>
> - (2 * α * β * 1): Decoding time of the Guiding model pair (the Guiding model participates in each decoding step).
> - (1 - Ω): Decoding time of the Guided model.
> - γ: Additional time overhead caused by switching between the Guiding model pair and the Guided model for inference.
> - α: The ratio of the parameter size of a single Guiding model to that of the Guided model.
> - β: The inference speed of the Guiding model relative to the Guided model at the same parameter size.
> - Ω: The average substitution ratio of decoding by the Guiding model pair.
>
> For example, in Gemma2-2b-it → Gemma2-27b:
>
> - β = 1,
> - α = 0.074,
> - Ω = 0.3,
> - γ is estimated as 0.422.
>
> This estimation indicates that the current implementation of GOOD can further improve its speed and achieve better inference performance than Vanilla Decoding by addressing the following directions:
>
> 1. Since the two Guiding models can execute in parallel, the decoding time of the Guiding model pair could potentially be reduced from 2 * α * β * 1 to α * β * 1.
> 2. Since communication between the Guiding models involves only string exchanges with minimal overhead, the Guiding model pair and the Guided model can be deployed separately to reduce the overhead of switching models, potentially significantly decreasing γ.
> 3. Since the Guiding models already perform predictions before the Guided model's inference, the Guiding models can be viewed as Speculative Inference SSMs, with the Guided model acting as the Verifier. According to estimates from SpecInfer[1], the decoding performance of the Guided model can potentially improve by 1.3–2.4× without additional overhead.
>
> Thus, under the most ideal implementation, the GOOD decoding performance could be optimized to:
>
> (α * β * 1) + (1 - Ω) * 0.42
>
> For Gemma2-2b-it → Gemma2-27b, this value equals 0.368.
>
> Although the current implementation is far from achieving this theoretical performance, we believe that GOOD can be further improved to achieve inference performance superior to Vanilla Decoding.
>
> [1] Miao, Xupeng, et al. "Specinfer: Accelerating large language model serving with tree-based speculative inference and verification." Proceedings of the 29th ACM International Conference on Architectural Support for Programming Languages and Operating Systems, Volume 3. 2024.

---

> ### Author Response · Authors · 2024-11-24
> **Response to Reviewer KAGp: W3, W4, Q3, Q4**
>
> ## W3, W4, Q3, Q4：
>
> Thank you for pointing out these issues. We have supplemented the relevant experimental data as follows:
>
> | Method | Model | MT-Bench | AlpacaEval | Harmless |
> | --- | --- | --- | --- | --- |
> | Vanilla (Baseline) | Gemma2-2b-it | 7.60 | 35.65 | 0.96 |
> | Vanilla (Baseline) | Gemma2-9b-it | 8.34 | 34.53 | 0.97 |
> | GOOD | Gemma2-2b-it → Gemma2-9b | 7.81 | 32.05 | 0.95 |
> | Proxy-Tuning | Gemma2-2b-it → Gemma2-9b | 3.81 | 9.94 | 0.90 |
> | GaC | Gemma2-2b-it + Gemma2-9b | 5.52 | 10.12 | 0.88 |
>
> The results indicate that GOOD achieves alignment performance close to direct fine-tuning across multiple benchmarks, while significantly outperforming the baseline methods Proxy-Tuning and GaC.

---

> > ### Comment · Reviewer_KAGp · 2024-11-27
> >
> > Thank you for your responses.
> >
> > I have some doubts regarding the results of proxy tuning, because the results shown in paper [1] indicate that proxy tuning performs well on Llama2, even achieving performance comparable to directly fine-tuned models on AlpacaEval. Why, then, does proxy tuning perform so poorly in your experiments?
> >
> > [1] Tuning Language Models by Proxy

---

> ### Author Response · Authors · 2024-11-24
> **Response to Reviewer KAGp:**
>
> ## w5, w6
> Regarding the formatting issues you pointed out in W5 and W6, I will make the necessary improvements in the revised version of the paper to be submitted shortly. Thank you for your understanding!

---

> ### Author Response · Authors · 2024-11-24
> **Response to Reviewer KAGp:**
>
> We have also added further explanations about the practical value of the GOOD method. We hope this provides you with a clearer understanding of the approach：
>
> ## Application Scenarios and Value of GOOD
>
> ### Reducing Repetitive Fine-tuning
>
> GOOD enables fine-tuning conducted on one LLM to be transferred to another LLM, thereby avoiding unnecessary repetitive fine-tuning and reducing the number of model variants caused by different fine-tuning processes. Even if these models differ only slightly, their redundant storage can lead to significant waste of storage resources.
>
> ### Studying the Impact of Fine-tuning
>
> GOOD can be used to analyze the sources of performance gains from fine-tuning. For example, it can help determine whether the performance improvement stems from changes in linguistic habits or from deeper learning. If it is the former, simply transferring these linguistic habits to other models may achieve similar performance gains. If it is the latter, the gains from such a transfer should be significantly lower than direct fine-tuning (note that this comparison should use equally accurate alignment discrimination).
>
> ### LLM Edge-Cloud Collaboration
>
> Since GOOD involves only string-level information exchange between the guiding model pair and the guided model, it enables low-cost collaboration between edge models and cloud models during decoding. Our analysis in Section 5.2 shows that GOOD-guided alignment retains the alignment characteristics of the guiding models. This means that in LLM edge-cloud collaboration supported by GOOD, it is possible to perform customized fine-tuning of edge models without exposing user-private conversational data. This allows the collaborative output to incorporate both user-specific customization and the powerful capabilities of cloud models.
>
> In this scenario, cloud models can use not only pretrained models but also aligned models. We conducted a series of tests demonstrating that in GOOD-supported edge-cloud collaboration, overall performance improves as the cloud model’s performance enhances, even without updating the user’s edge model:
>
> | Model | MT-Bench Score |
> | --- | --- |
> | Gemma2-2b-it | 7.60 |
> | Llama3-8b-Instruct | 7.75 |
> | Gemma2-2b-it→Llama3-8b-Instruct | 7.33 |
> | Qwen2-7b-Instruct | 8.02 |
> | Gemma2-2b-it→Qwen2-7b-Instruct | 7.80 |
> | Gemma2-9b-it | 8.34 |
> | Gemma2-2b-it→Gemma2-9b-it | 8.44 |
>
> This characteristic means that GOOD can extend the lifespan of customized models by preserving the unique features of various local fine-tunings while keeping their performance up-to-date, rather than allowing them to quickly fall behind newer models and require frequent updates.
>
> ### GOOD as Further Validation of the Superficial Alignment Hypothesis
>
> The Superficial Alignment Hypothesis suggests that most of a model’s knowledge and capabilities are acquired during pretraining, with alignment primarily teaching the model which sub-distribution of responses to utilize in user interactions. The GOOD method replaces part of the decoding results of one model with alignment-related tokens from another model, without any fine-tuning, yet effectively achieves alignment. This supports the Superficial Alignment Hypothesis to some extent, indicating that alignment in models likely changes linguistic habits rather than learning new knowledge or capabilities.

---

> ### Author Response · Authors · 2024-11-24
> **Response to Reviewer KAGp:**
>
> We sincerely appreciate Reviewer KAGp's insightful suggestions and the opportunity to address the points raised. Your feedback has been instrumental in guiding our revisions and enhancing the clarity and robustness of our work.
>
> In response to your comments, we have conducted additional experiments, refined our analyses, and provided further discussions on the theoretical and practical implications of our approach. These efforts reflect our commitment to addressing the issues you identified and ensuring the manuscript meets the highest standards.
>
> We are currently incorporating these changes into the revised manuscript and plan to submit the updated version shortly. We hope the results and discussions provided here demonstrate the potential and significance of the GOOD method, as well as our dedication to improving the quality of our work.
>
> We hope that our responses and the additional experiments and analyses provided above address your concerns and align with your expectations, and we would deeply value your consideration of their impact on your evaluation of the paper. Should there be any remaining points or further clarifications required, we are more than happy to address them. Once again, we sincerely thank you for your valuable feedback and guidance throughout this process.

---

> ### Author Response · Authors · 2024-11-25
> **Response to Reviewer KAGp: W2 (additional)**
>
> ## W2
>
> We conducted an evaluation comparing the decoding performance of three methods: GOOD (Gemma-2-9b-it → Gemma-2-27b), Proxy-Tuning (Gemma-2-9b-it → Gemma-2-27b), and Vanilla (Gemma-2-27b-it).
>
> The first question from MT-Bench was used as the prompt, with the maximum generation length set to 512. The experiments were performed on L40s (48G × 8).
>
> The results show that the decoding speeds of GOOD and Vanilla are unaffected by generation length, whereas Proxy-Tuning’s decoding speed slows down as the generation length increases (which might be due to specific implementation details— in theory, Proxy-Tuning could also maintain a decoding speed independent of generation length).
>
> For GOOD, the decoding speed exhibits two distinct regions: one where the guided model decoding is skipped (denoted as Region A) and another where skipping does not occur (denoted as Region B). Region A demonstrates significantly faster decoding compared to Vanilla decoding. When Speculative Inference is incorporated, Region B can also be accelerated, bringing its performance closer to or even surpassing Vanilla decoding. This suggests that further improvements to GOOD are very likely to achieve overall performance superior to Vanilla decoding, not only in terms of speed but also with lower computational costs.
>
> We will include the detailed charts in the revised version of the paper to be submitted soon. We hope this will make our explanation of GOOD’s inference performance clearer. Thank you for your valuable guidance!

---

> ### Author Response · Authors · 2024-11-25
> **Response to Reviewer KAGp's Comments and Revised Manuscript Submission**
>
> We sincerely thank you for your thoughtful and constructive feedback on our work. Your comments have been invaluable in helping us improve the quality and clarity of our work.
>
> In response to your suggestions, we have carefully revised the manuscript and submitted an updated version. Below, we summarize the key changes:
>
> 1. We have added comparisons and citations for relevant works, including DeAL, DeRa, EFT and so on.
> 2. The explanation of the GOOD algorithm has been improved, with additional details provided in the **Details of Alignment Discrimination** and **Implementation of Guidance** sections.
> 3. We updated the tables to use the three-line table format for better readability.
> 4. All figures have been checked and replaced with PDF versions to ensure they can be zoomed in for detail.
> 5. Additional discussions on the interference performance of GOOD have been included (Appendix D).
> 6. We have introduced a description of potential application scenarios for GOOD (Appendix E).
> 7. An analysis of GOOD's requirements for guiding model size has been added (Appendix F).
> 8. Analyses of the importance of **Alignment Discrimination** and **Token Substitution** are now included (Appendix G).
> 9. More comprehensive comparative results with baseline methods have been added (Appendix H).
>
> We hope these revisions address your concerns and enhance the manuscript’s overall quality and contribution.
>
> If you believe our revisions have resolved the issues raised in your review, we would greatly appreciate your reconsideration of the manuscript’s evaluation. If there are any remaining concerns or further suggestions for improvement, we would be more than happy to address them.
>
> Once again, we sincerely thank you for your time and effort in reviewing our work. Your feedback has been instrumental in shaping this improved version of our manuscript.

---

### Official Review · Reviewer_wSaE · 2024-10-28

**Soundness:** 2
**Presentation:** 3
**Contribution:** 2
**Rating:** 5
**Confidence:** 3

**Summary:**

This paper proposes a decoding-time LLM alignment method called GOOD (Guided Online Optimal Decoding), which utilizes a pair of guiding models (one aligned, one unaligned) to guide another model. Performing the alignment at decoding time is presented as an alternative to (resource-intensive) alignment during training (e.g., via SFT or RLHF). GOOD differs from other decoding-time alignment algorithms in that it does not require access to the guided model's parameters or vocabulary, and thus can be applied even to black-box models. GOOD is a two-step method: 1) Alignment discrimination, to identify positions which need alignment by comparing the logits of the two guiding models, and 2) Output substitution, where the guiding model's output replaces that of the guided model at the positions identified for alignment. This paper evaluates GOOD on MTBench and on the Helpful and Harmless dataset, and shows that it can achieve performance comparable to direct finetuning. Even when using guiding models which are weaker than the guided model, most of the performance of direct finetuning can be recovered at decoding time. The paper also shows that GOOD can further improve performance of already-aligned models. Finally, the paper performs some analyses to understand the importance of alignment discrimination (step 1 in the GOOD process) and its sensitivity to hyperparameter settings, and to understand similarity of GOOD-decoded outputs to both those of the guiding model and to those of the same model aligned directly through finetuning.

**Strengths:**

- The motivation is clear, including the relationship of GOOD to other training-time and post-training alignment methods.
- The results are compelling, especially those showing that 1) GOOD can perform on par with alignment via direct finetuning, and 2) most of these gains can still be preserved even when using a guiding model which is weaker than the guided model.
- There are some insightful ablations, including the ablation using random guidance (for understanding where the performance improvement from GOOD comes from), and the sweep over different guided decoding ratios (to understand the sensitivity of performance to the amount of alignment guidance provided).

**Weaknesses:**

- There is no mention of the inference-time cost of GOOD (Algorithm 1). The paper motivates decoding-time alignment methods by citing the resource-intensiveness of training-time approaches, but does not provide an analysis of the time complexity of GOOD or report results on how much it slows down inference (for different guiding models). The tradeoff between cost at training versus inference time is an important consideration.
- The description of the GOOD algorithm in Figure 1 and Algorithm 1 (and in the text) is underspecified and missing key details. See specific questions in "Questions" section.
- The claims in the paper regarding where the GOOD performance enhancement comes from are not fully substantiated. See specific questions in "Questions" section.

**Questions:**

- Questions regarding the GOOD algorithm (Algorithm 1):
  - At every iteration of the while loop, are you re-decoding from the start of the input sequence I_{B}? If a token(s) is replaced in I_{B}, are the remaining decoding steps using model B conditioned on these replaced tokens?
  - Given that I_{A}, I_{A_{it}}, and I_{B} are not (necessarily) equal, the decodes t_{A}, t_{A_{it}}, and t_{B} are conditioned on different prefixes, and it seems likely they'd often be misaligned (e.g. different word order, etc). So is there any guarantee that i) the comparison of logits t_{A} and t_{A_{it}} (step 6) make sense, and 2) the replacement token t_{A_{it}} (step 8) makes sense in the context of the output O_B?
- The paper mentions the importance of the alignment discrimination step, but only Max Match and a variant of Top P Overlap were considered. Were any ablations performed to investigate the effect of the discrimination method? Were alternative alignment discrimination methods (e.g. measures of distance between probability distributions) considered?
- The paper states that the performance improvement from GOOD "mainly stems from accurately identifying positions that need alignment". Figure 5 shows that random guidance substantially underperforms guided decoding (with a variant of Top P Overlap), but does not take into account the quality of the guiding model. Table 1, on the other hand, shows the performance of GOOD for a few different guiding models, but does not report the performance of the guiding model itself. Is there any complete analysis showing the relative importance of step 1 (alignment discrimination) versus step 2 (output substitution)?
- There is no discussion of how the A_{it} guiding models used in the experiments are aligned. Is it via SFT and/or RLHF, and which datasets are used to perform the alignment?

Nits:
- Figure 2 and Figure 3 are small and hard to read.
- The fact that GOOD doesn't require access to the guided model's parameters is emphasized several times, but there is no mention that access to the logits of the *guiding* models is indeed required. For clarity, it would probably be worth mentioning this point.
- Also, it is emphasized that access to the guided model's vocabulary is not required, but access to the guiding model's tokenizer is in fact required. This could be misleading, and should probably be clarified.

---

> ### Author Response · Authors · 2024-11-24
> **Response to Reviewer wSaE: w1**
>
> We sincerely appreciate Reviewer wSaE's time, effort, insightful feedback, and constructive suggestions, which are valuable in helping us refine our work. Below, we address the concerns raised and provide additional discussions and justifications to clarify our contribution.
>
> ## W1：
>
> We conducted tests based on the current implementation of GOOD and analyzed the theoretical optimal performance of the method.
>
> ### Current Performance
>
> We evaluated the alignment of Gemma2-2b-it → Gemma2-27b and Gemma2-2b-it → Qwen2-72. The detailed test configurations are as follows:
>
> - Due to varying GPU memory requirements for different configurations, the speed measurements for Gemma2-2b-it → Gemma2-27b and Gemma2-27b-it were conducted on L40s (48GB × 8), while the speed measurements for Gemma2-2b-it → Qwen2-72 and Qwen2-72-Instruct were performed on A100 (80GB × 8).
> - For Gemma2-27b-it and Qwen2-72-Instruct, generation was conducted using the Huggingface Transformers library.
> - The test question set was sourced from MT-Bench, covering multiple question categories.
> - Model inference utilized caching, and all models involved were deployed using model parallelism.
> - We examined the decoding speed under various configurations as a function of generation length and confirmed that the number of tokens already generated had no significant impact on model inference speed.
>
> The test results show that the average decoding speed for Gemma2-2b-it → Gemma2-27b is 1.27 times that of Gemma2-27b-it, and the average decoding speed for Gemma2-2b-it → Qwen2-72 is 1.15 times that of Qwen2-72-Instruct.
>
> ### Optimization Potential
>
> It is worth noting that the current implementation of GOOD still holds significant potential for performance improvement. In the current implementation, for each token's decoding, the Guiding model pair is first inferred, and based on its judgment, it is determined whether the Guided model needs to be inferred (if alignment is deemed unnecessary, the Guided model's inference is skipped). The estimated time complexity formula for the current implementation is given below (taking the inference speed of the original model as the baseline value of 1):
>
> T(GOOD)/T(Vanilla) = (2 * α * β * 1) + (1 - Ω) + γ
>
> The formula and symbol definitions are as follows:
>
> - (2 * α * β * 1): Decoding time of the Guiding model pair (the Guiding model participates in each decoding step).
> - (1 - Ω): Decoding time of the Guided model.
> - γ: Additional time overhead caused by switching between the Guiding model pair and the Guided model for inference.
> - α: The ratio of the parameter size of a single Guiding model to that of the Guided model.
> - β: The inference speed of the Guiding model relative to the Guided model at the same parameter size.
> - Ω: The average substitution ratio of decoding by the Guiding model pair.
>
> For example, in Gemma2-2b-it → Gemma2-27b:
>
> - β = 1,
> - α = 0.074,
> - Ω = 0.3,
> - γ is estimated as 0.422.
>
> This estimation indicates that the current implementation of GOOD can further improve its speed and achieve better inference performance than Vanilla Decoding by addressing the following directions:
>
> 1. Since the two Guiding models can execute in parallel, the decoding time of the Guiding model pair could potentially be reduced from 2 * α * β * 1 to α * β * 1.
> 2. Since communication between the Guiding models involves only string exchanges with minimal overhead, the Guiding model pair and the Guided model can be deployed separately to reduce the overhead of switching models, potentially significantly decreasing γ.
> 3. Since the Guiding models already perform predictions before the Guided model's inference, the Guiding models can be viewed as Speculative Inference SSMs, with the Guided model acting as the Verifier. According to estimates from SpecInfer[1], the decoding performance of the Guided model can potentially improve by 1.3–2.4× without additional overhead.
>
> Thus, under the most ideal implementation, the GOOD decoding performance could be optimized to:
>
> (α * β * 1) + (1 - Ω) * 0.42
>
> For Gemma2-2b-it → Gemma2-27b, this value equals 0.368.
>
> Although the current implementation is far from achieving this theoretical performance, we believe that GOOD can be further improved to achieve inference performance superior to Vanilla Decoding.
>
> [1] Miao, Xupeng, et al. "Specinfer: Accelerating large language model serving with tree-based speculative inference and verification." Proceedings of the 29th ACM International Conference on Architectural Support for Programming Languages and Operating Systems, Volume 3. 2024.

---

> ### Author Response · Authors · 2024-11-24
> **Response to Reviewer wSaE:**
>
> ## W2, Q1：
>
> Thank you very much for pointing out these issues! We will address them in the revised version of the paper and provide more detailed explanations.
>
> At every iteration of the while loop, we do not decode \( I_B \) from scratch but rather perform incremental decoding. Since the vocabulary of the guiding model and the guided model may differ, a single token generated by the guiding model might be interpreted as multiple tokens by the guided model. In such cases, we treat it as the generation of multiple tokens and directly concatenate them to the `input_ids` of the guided model. Therefore, there is no token misalignment issue here.
>
> Although the tokenization of the same sentence might differ between the guiding model and the guided model, our algorithm consistently ensures that the strings received by both models remain synchronized. This guarantees that the discrimination and replacement of \( \text{logits } t_A \) and \( \text{logits } t_{A_{\text{it}}} \) are meaningful within the context of the output \( O_B \).

---

> ### Author Response · Authors · 2024-11-24
> **Response to Reviewer wSaE: W3, Q3**
>
> ## W3, Q3：
>
> In response to the issues you raised, we have conducted additional experiments, and the results are reported below:
>
> ### (1) Considering Different Guiding Models
>
> | Model | MT-Bench Score | Guided Ratio |
> | --- | --- | --- |
> | Llama2-13b-chat → gemma2_9b | 5.57 | 0.19 |
> | Mistral-7B-Instruct-v0.3 → gemma2_9b | 6.84 | 0.16 |
> | Qwen2-7b-Instruct → gemma2_9b | 7.39 | 0.23 |
>
> These results indicate that even when using guiding models of varying quality, performance consistently surpasses random replacement decoding (Guided Ratio = 0.3), even at lower decoding ratios.
>
> ### (2) Quality of Guiding Models Used in Table 1
>
> | Model | Harmless Ratio |
> | --- | --- |
> | Gemma-2b-it | (under testing) |
> | Gemma-7b-it | (under testing) |
> | Gemma2-2b-it | 0.963 |
> | Gemma2-9b-it | 0.974 |

---

> ### Author Response · Authors · 2024-11-24
> **Response to Reviewer wSaE:**
>
> We have also added further explanations about the practical value of the GOOD method. We hope this provides you with a clearer understanding of the approach：
>
> ## Application Scenarios and Value of GOOD
>
> ### Reducing Repetitive Fine-tuning
>
> GOOD enables fine-tuning conducted on one LLM to be transferred to another LLM, thereby avoiding unnecessary repetitive fine-tuning and reducing the number of model variants caused by different fine-tuning processes. Even if these models differ only slightly, their redundant storage can lead to significant waste of storage resources.
>
> ### Studying the Impact of Fine-tuning
>
> GOOD can be used to analyze the sources of performance gains from fine-tuning. For example, it can help determine whether the performance improvement stems from changes in linguistic habits or from deeper learning. If it is the former, simply transferring these linguistic habits to other models may achieve similar performance gains. If it is the latter, the gains from such a transfer should be significantly lower than direct fine-tuning (note that this comparison should use equally accurate alignment discrimination).
>
> ### LLM Edge-Cloud Collaboration
>
> Since GOOD involves only string-level information exchange between the guiding model pair and the guided model, it enables low-cost collaboration between edge models and cloud models during decoding. Our analysis in Section 5.2 shows that GOOD-guided alignment retains the alignment characteristics of the guiding models. This means that in LLM edge-cloud collaboration supported by GOOD, it is possible to perform customized fine-tuning of edge models without exposing user-private conversational data. This allows the collaborative output to incorporate both user-specific customization and the powerful capabilities of cloud models.
>
> In this scenario, cloud models can use not only pretrained models but also aligned models. We conducted a series of tests demonstrating that in GOOD-supported edge-cloud collaboration, overall performance improves as the cloud model’s performance enhances, even without updating the user’s edge model:
>
> | Model | MT-Bench Score |
> | --- | --- |
> | Gemma2-2b-it | 7.60 |
> | Llama3-8b-Instruct | 7.75 |
> | Gemma2-2b-it→Llama3-8b-Instruct | 7.33 |
> | Qwen2-7b-Instruct | 8.02 |
> | Gemma2-2b-it→Qwen2-7b-Instruct | 7.80 |
> | Gemma2-9b-it | 8.34 |
> | Gemma2-2b-it→Gemma2-9b-it | 8.44 |
>
> This characteristic means that GOOD can extend the lifespan of customized models by preserving the unique features of various local fine-tunings while keeping their performance up-to-date, rather than allowing them to quickly fall behind newer models and require frequent updates.
>
> ### GOOD as Further Validation of the Superficial Alignment Hypothesis
>
> The Superficial Alignment Hypothesis suggests that most of a model’s knowledge and capabilities are acquired during pretraining, with alignment primarily teaching the model which sub-distribution of responses to utilize in user interactions. The GOOD method replaces part of the decoding results of one model with alignment-related tokens from another model, without any fine-tuning, yet effectively achieves alignment. This supports the Superficial Alignment Hypothesis to some extent, indicating that alignment in models likely changes linguistic habits rather than learning new knowledge or capabilities.

---

> ### Author Response · Authors · 2024-11-24
> **Response to Reviewer wSaE:**
>
> We sincerely appreciate Reviewer wSaE's constructive feedback and thoughtful suggestions, which have been instrumental in helping us refine and improve our work. In response to your insightful comments, we have conducted extensive additional experiments, enhanced theoretical discussions, and provided more comprehensive explanations to address the concerns raised.
>
> We are currently conducting additional experiments for Q2, Q3 (3), and Q4 and are incorporating these results into the revised manuscript. If you believe our efforts and revisions have addressed your concerns effectively, we would greatly appreciate your further consideration of our work.
>
> Once again, we are deeply grateful for your valuable input, which has significantly contributed to enhancing the quality and depth of our research. Please feel free to let us know if there are any additional points requiring further clarification or refinement. Thank you for your guidance and support!

---

> ### Author Response · Authors · 2024-11-25
> **Response to Reviewer wSaE: Q2**
>
> ## Q2
>
> The issue you mentioned is indeed crucial. Our experimental analysis also highlights the critical importance of the alignment discrimination step in the GOOD method. However, our current methods, such as Max Match and Top P Overlap, are quite basic and intuitive. Improving these methods is certainly necessary. We believe that enhancing this aspect could significantly boost the performance of the GOOD method. Currently, we have not yet explored this direction and plan to leave it as part of our future work.

---

> ### Author Response · Authors · 2024-11-25
> **Response to Reviewer wSaE: Q3(3)**
>
> ## Q3(3)
>
> We have compiled the following data and conducted additional experiments to address your question.
>
> **Importance of Alignment Discrimination: More Accurate Identification Leads to Better Effectiveness**
>
> | Model | MT-Bench Score |
> | --- | --- |
> | Gemma-2-2b-it → Gemma-2-9b (**with AD**) | 8.13 |
> | Gemma-2-2b-it → Gemma-2-9b | 7.81 |
> | Gemma-2-2b-it → Qwen2-7b (**with AD**) | 8.09 |
> | Gemma-2-2b-it → Qwen2-7b | 7.35 |
>
> *(In rows labeled **with AD**, the most accurate alignment discrimination was used. This means comparing the aligned version of the guided model with the unaligned version (e.g., Gemma-2-9b-it vs. Gemma-2-9b) to determine whether alignment is needed. If alignment is required, the guiding model’s output is used at that position.)*
>
> **Importance of Output Replacement: Stronger Guidance Yields Better Results**
>
> | Model | MT-Bench Score |
> | --- | --- |
> | Gemma-2-9b-it → Qwen2-72B | 8.12 |
> | Qwen2-7B-Instruct → Qwen2-72B | 8.38 |
> | Qwen2-7B-Instruct (discrimination) + Gemma-2-9b-it (guidance) → Qwen2-72B | 8.45 |
>
> **Even with the Most Accurate Alignment Discrimination, the Quality of the Guiding Model Affects Final Performance**
>
> | Model | MT-Bench Score |
> | --- | --- |
> | Gemma-2-2b-it | 7.60 |
> | Qwen2.5-0.5b-Instruct | 5.01 |
> | Gemma-2-2b-it → Gemma-2-9b (**with AD**) | 8.13 |
> | Qwen2.5-0.5b-Instruct → Gemma-2-9b (**with AD**) | 6.70 |
> | Gemma-2-2b-it → Qwen2-7b (**with AD**) | 8.03 |
> | Qwen2.5-0.5b-Instruct → Qwen2-7b (**with AD**) | 7.19 |
>
> These results highlight that both the accuracy of alignment discrimination and the quality of the guiding model are crucial for achieving optimal performance. Even with the most accurate alignment discrimination, the quality of the guiding model plays a significant role in determining the final outcomes.

---

> ### Author Response · Authors · 2024-11-25
> **Response to Reviewer wSaE: w1 (additional)**
>
> ## W1
>
> We conducted an evaluation comparing the decoding performance of three methods: GOOD (Gemma-2-9b-it → Gemma-2-27b), Proxy-Tuning (Gemma-2-9b-it → Gemma-2-27b), and Vanilla (Gemma-2-27b-it).
>
> The first question from MT-Bench was used as the prompt, with the maximum generation length set to 512. The experiments were performed on L40s (48G × 8).
>
> The results show that the decoding speeds of GOOD and Vanilla are unaffected by generation length, whereas Proxy-Tuning’s decoding speed slows down as the generation length increases (which might be due to specific implementation details— in theory, Proxy-Tuning could also maintain a decoding speed independent of generation length).
>
> For GOOD, the decoding speed exhibits two distinct regions: one where the guided model decoding is skipped (denoted as Region A) and another where skipping does not occur (denoted as Region B). Region A demonstrates significantly faster decoding compared to Vanilla decoding. When Speculative Inference is incorporated, Region B can also be accelerated, bringing its performance closer to or even surpassing Vanilla decoding. This suggests that further improvements to GOOD are very likely to achieve overall performance superior to Vanilla decoding, not only in terms of speed but also with lower computational costs.
>
> We will include detailed charts in the revised version of the paper to be submitted soon. We hope this will make our explanation of GOOD’s inference performance clearer. Thank you for your valuable guidance!

---

> ### Author Response · Authors · 2024-11-25
> **Response to Reviewer wSaE's Comments and Revised Manuscript Submission**
>
> We sincerely thank you for your thoughtful and constructive feedback on our work. Your comments have been invaluable in helping us improve the quality and clarity of our work.
>
> In response to your suggestions, we have carefully revised the manuscript and submitted an updated version. Below, we summarize the key changes:
>
> 1. We have added comparisons and citations for relevant works, including DeAL, DeRa, EFT and so on.
> 2. The explanation of the GOOD algorithm has been improved, with additional details provided in the **Details of Alignment Discrimination** and **Implementation of Guidance** sections.
> 3. We updated the tables to use the three-line table format for better readability.
> 4. All figures have been checked and replaced with PDF versions to ensure they can be zoomed in for detail.
> 5. Additional discussions on the interference performance of GOOD have been included (Appendix D).
> 6. We have introduced a description of potential application scenarios for GOOD (Appendix E).
> 7. An analysis of GOOD's requirements for guiding model size has been added (Appendix F).
> 8. Analyses of the importance of **Alignment Discrimination** and **Token Substitution** are now included (Appendix G).
> 9. More comprehensive comparative results with baseline methods have been added (Appendix H).
>
> We hope these revisions address your concerns and enhance the manuscript’s overall quality and contribution.
>
> If you believe our revisions have resolved the issues raised in your review, we would greatly appreciate your reconsideration of the manuscript’s evaluation. If there are any remaining concerns or further suggestions for improvement, we would be more than happy to address them.
>
> Once again, we sincerely thank you for your time and effort in reviewing our work. Your feedback has been instrumental in shaping this improved version of our manuscript.

---

> > ### Comment · Reviewer_wSaE · 2024-11-27
> >
> > Thank you for the additional results and discussion, which addressed some of my questions. The high inference-time costs currently observed with your method could still hinder its practical applicability (and the proposed optimizations have not yet been implemented/verified). The contribution of this paper relative to similar methods such as proxy-tuning is also not convincing, and the story/organization around where the performance improvement from GOOD comes from still needs to be clarified (especially with respect to the relative importance of 1) alignment discrimination method and 2) guiding model quality).

---

### Official Review · Reviewer_Yvpt · 2024-11-01

**Soundness:** 3
**Presentation:** 3
**Contribution:** 2
**Rating:** 5
**Confidence:** 3

**Summary:**

This work proposes GOOD, a decoding-time alignment method that aligns raw pre-trained LMs without tuning. Specifically, a pair of Base and Instruction guiding LMs are utilized to identify the timing where guidance is needed, if this is the case, then the predictions from the guiding Instruction model is appended rather than the token decoded from the original model. Experiments in varioius evaluation settings are conducted, where the results indicate the effectiveness of the proposed method.

**Strengths:**

- Decoding-time alignment is an interesting research direction, which alleviate the potential costly tuning of LMs.
- The proposed method can obtain reasonable results, and the analyses provide some interesting insights.

**Weaknesses:**

- This method still requires an aligned guiding model, which is still the resulting model from tuning-based alignment.
- Some parts of the algorithm is not well explained, specifically, it seems to be an important decision of when alignment is needed. More details on this part should be provided.

**Questions:**

- The idea is in some way very similar to the one proposed by Mitchell et al. (2023), comparisons or at least discussions should be provided.

[1] An Emulator for Fine-Tuning Large Language Models using Small Language Models, Mitchell et al., 2023

---

> ### Author Response · Authors · 2024-11-24
> **Response to Reviewer Yvpt: w1, w2, q1**
>
> We sincerely appreciate Reviewer Yvpt's time, effort, insightful feedback, and constructive suggestions, which are valuable in helping us refine our work. Below, we address the concerns raised and provide additional discussions and justifications to clarify our contribution.
>
> ## W1：
>
> You are absolutely correct in pointing out that GOOD still relies on alignment models based on fine-tuning. We truly appreciate your insightful observation. However, we believe this reliance can also be seen as a potential strength. It allows GOOD to benefit from the effectiveness of existing, well-established alignment methods. By doing so, it helps minimize unnecessary fine-tuning efforts while achieving results comparable to direct fine-tuning, even without performing fine-tuning itself.
>
> ## W2：
>
> Thank you for pointing out this issue. We will revise the paper and provide a more detailed explanation in the methodology section.
>
> GOOD employs a pair of guiding models: one is a pretrained model, and the other is its aligned version. During inference, the token logits generated by these two models exhibit some differences. By evaluating the extent of this token difference, we determine whether alignment is needed at the current position (the paper mentions methods such as Max Match and Top_p overlap as examples of such discrimination). If alignment is deemed necessary, the output from the aligned version of the guiding model pair is used; otherwise, the output from the original model is used. This decision-making happens at each token decoding step after the guiding model pair completes inference.
>
> The criteria for determining whether alignment is needed are diverse and can be tailored to specific tasks. For the logits (predicted probability distribution of the next token) generated by model A and model A_it, the following methods can be used to decide whether to align:
>
> Max Match: This method checks if the most probable token predicted by model A matches that of model A_it. If they differ, it is inferred that alignment is needed.
>
> Top P Overlap: This method measures the overlap between the Top P tokens of the two distributions, where Top P tokens are those with the highest probabilities summing to a cumulative probability P. For example, if the Top P tokens of model A share less than a certain threshold proportion of tokens with model A_it, alignment is triggered.
>
> Top K Overlap: This method compares the Top K tokens (those with the K highest individual probabilities) of the two distributions. If the overlap ratio falls below a threshold, alignment is deemed necessary.
>
> These methods can be adjusted using hyperparameters, such as the value of P or K, or the overlap thresholds, to control the sensitivity of alignment. For instance, in tasks where high precision is critical, a stricter threshold can be applied.
>
> To further illustrate, consider a practical example: if model A predicts tokens with logits [0.6, 0.3, 0.1] for tokens [t_1, t_2, t_3], and model A_it predicts logits [0.4, 0.5, 0.1] for the same tokens, the most probable token differs (t_1 for A, t_2 for A_it). Here, alignment would be triggered under the Max Match criterion.
>
> The guided model can either wait until the guiding model pair completes inference or decode simultaneously, depending on the implementation.
>
> ## Q1：
>
> Thank you for providing this reference paper-it is very useful for enriching the background section of our work. We will include a citation to this paper in the revised version of the manuscript.
>
> After carefully reviewing the paper, we find that the proposed method is very similar to proxy-tuning. In addition to basic inference, the authors propose combining this method with speculative decoding to accelerate execution, using a smaller model to predict the decoding of a larger pretrained model. However, this approach shares a similar limitation: the guiding model and the guided model must use the same vocabulary.
>
> Our method also considers acceleration. When the discriminator determines that alignment is needed, the inference of the guided model can be skipped, thereby reducing the inference time at that position. In GOOD, the mechanism of Speculative Inference can be further integrated into the current acceleration method to achieve additional speed-up. For instance, at positions that do not require alignment, guiding models could make early predictions while the guided model acts as a verifier. At positions requiring alignment, the guided model could skip inference entirely. Moreover, our method offers greater flexibility since the guiding model pair and the guided model only need to exchange information at the string level.

---

> ### Author Response · Authors · 2024-11-24
> **Response to Reviewer Yvpt:**
>
> We have also added further explanations about the computational cost and practical value of the GOOD method. We hope this provides you with a clearer understanding of the approach：
>
> ## 1. Computational Cost and Optimization Potential of GOOD
>
> We conducted tests based on the current implementation of GOOD and analyzed the theoretical optimal performance of the method.
>
> ### Current Performance
>
> We evaluated the alignment of Gemma2-2b-it → Gemma2-27b and Gemma2-2b-it → Qwen2-72. The detailed test configurations are as follows:
>
> - Due to varying GPU memory requirements for different configurations, the speed measurements for Gemma2-2b-it → Gemma2-27b and Gemma2-27b-it were conducted on L40s (48GB × 8), while the speed measurements for Gemma2-2b-it → Qwen2-72 and Qwen2-72-Instruct were performed on A100 (80GB × 8).
> - For Gemma2-27b-it and Qwen2-72-Instruct, generation was conducted using the Huggingface Transformers library.
> - The test question set was sourced from MT-Bench, covering multiple question categories.
> - Model inference utilized caching, and all models involved were deployed using model parallelism.
> - We examined the decoding speed under various configurations as a function of generation length and confirmed that the number of tokens already generated had no significant impact on model inference speed.
>
> The test results show that the average decoding speed for Gemma2-2b-it → Gemma2-27b is 1.27 times that of Gemma2-27b-it, and the average decoding speed for Gemma2-2b-it → Qwen2-72 is 1.15 times that of Qwen2-72-Instruct.
>
> ### Optimization Potential
>
> It is worth noting that the current implementation of GOOD still holds significant potential for performance improvement. In the current implementation, for each token's decoding, the Guiding model pair is first inferred, and based on its judgment, it is determined whether the Guided model needs to be inferred (if alignment is deemed unnecessary, the Guided model's inference is skipped). The estimated time complexity formula for the current implementation is given below (taking the inference speed of the original model as the baseline value of 1):
>
> T(GOOD)/T(Vanilla) = (2 * α * β * 1) + (1 - Ω) + γ
>
> The formula and symbol definitions are as follows:
>
> - (2 * α * β * 1): Decoding time of the Guiding model pair (the Guiding model participates in each decoding step).
> - (1 - Ω): Decoding time of the Guided model.
> - γ: Additional time overhead caused by switching between the Guiding model pair and the Guided model for inference.
> - α: The ratio of the parameter size of a single Guiding model to that of the Guided model.
> - β: The inference speed of the Guiding model relative to the Guided model at the same parameter size.
> - Ω: The average substitution ratio of decoding by the Guiding model pair.
>
> For example, in Gemma2-2b-it → Gemma2-27b:
>
> - β = 1,
> - α = 0.074,
> - Ω = 0.3,
> - γ is estimated as 0.422.
>
> This estimation indicates that the current implementation of GOOD can further improve its speed and achieve better inference performance than Vanilla Decoding by addressing the following directions:
>
> 1. Since the two Guiding models can execute in parallel, the decoding time of the Guiding model pair could potentially be reduced from 2 * α * β * 1 to α * β * 1.
> 2. Since communication between the Guiding models involves only string exchanges with minimal overhead, the Guiding model pair and the Guided model can be deployed separately to reduce the overhead of switching models, potentially significantly decreasing γ.
> 3. Since the Guiding models already perform predictions before the Guided model's inference, the Guiding models can be viewed as Speculative Inference SSMs, with the Guided model acting as the Verifier. According to estimates from SpecInfer[1], the decoding performance of the Guided model can potentially improve by 1.3–2.4× without additional overhead.
>
> Thus, under the most ideal implementation, the GOOD decoding performance could be optimized to:
>
> (α * β * 1) + (1 - Ω) * 0.42
>
> For Gemma2-2b-it → Gemma2-27b, this value equals 0.368.
>
> Although the current implementation is far from achieving this theoretical performance, we believe that GOOD can be further improved to achieve inference performance superior to Vanilla Decoding.
>
> [1] Miao, Xupeng, et al. "Specinfer: Accelerating large language model serving with tree-based speculative inference and verification." Proceedings of the 29th ACM International Conference on Architectural Support for Programming Languages and Operating Systems, Volume 3. 2024.

---

> ### Author Response · Authors · 2024-11-24
> **Response to Reviewer Yvpt:**
>
> ## 2. Application Scenarios and Value of GOOD
>
> ### Reducing Repetitive Fine-tuning
>
> GOOD enables fine-tuning conducted on one LLM to be transferred to another LLM, thereby avoiding unnecessary repetitive fine-tuning and reducing the number of model variants caused by different fine-tuning processes. Even if these models differ only slightly, their redundant storage can lead to significant waste of storage resources.
>
> ### Studying the Impact of Fine-tuning
>
> GOOD can be used to analyze the sources of performance gains from fine-tuning. For example, it can help determine whether the performance improvement stems from changes in linguistic habits or from deeper learning. If it is the former, simply transferring these linguistic habits to other models may achieve similar performance gains. If it is the latter, the gains from such a transfer should be significantly lower than direct fine-tuning (note that this comparison should use equally accurate alignment discrimination).
>
> ### LLM Edge-Cloud Collaboration
>
> Since GOOD involves only string-level information exchange between the guiding model pair and the guided model, it enables low-cost collaboration between edge models and cloud models during decoding. Our analysis in Section 5.2 shows that GOOD-guided alignment retains the alignment characteristics of the guiding models. This means that in LLM edge-cloud collaboration supported by GOOD, it is possible to perform customized fine-tuning of edge models without exposing user-private conversational data. This allows the collaborative output to incorporate both user-specific customization and the powerful capabilities of cloud models.
>
> In this scenario, cloud models can use not only pretrained models but also aligned models. We conducted a series of tests demonstrating that in GOOD-supported edge-cloud collaboration, overall performance improves as the cloud model’s performance enhances, even without updating the user’s edge model:
>
> | Model | MT-Bench Score |
> | --- | --- |
> | Gemma2-2b-it | 7.60 |
> | Llama3-8b-Instruct | 7.75 |
> | Gemma2-2b-it→Llama3-8b-Instruct | 7.33 |
> | Qwen2-7b-Instruct | 8.02 |
> | Gemma2-2b-it→Qwen2-7b-Instruct | 7.80 |
> | Gemma2-9b-it | 8.34 |
> | Gemma2-2b-it→Gemma2-9b-it | 8.44 |
>
> This characteristic means that GOOD can extend the lifespan of customized models by preserving the unique features of various local fine-tunings while keeping their performance up-to-date, rather than allowing them to quickly fall behind newer models and require frequent updates.
>
> ### GOOD as Further Validation of the Superficial Alignment Hypothesis
>
> The Superficial Alignment Hypothesis suggests that most of a model’s knowledge and capabilities are acquired during pretraining, with alignment primarily teaching the model which sub-distribution of responses to utilize in user interactions. The GOOD method replaces part of the decoding results of one model with alignment-related tokens from another model, without any fine-tuning, yet effectively achieves alignment. This supports the Superficial Alignment Hypothesis to some extent, indicating that alignment in models likely changes linguistic habits rather than learning new knowledge or capabilities.

---

> ### Author Response · Authors · 2024-11-24
> **Response to Reviewer Yvpt:**
>
> We sincerely appreciate Reviewer Yvpt's valuable suggestions and insightful feedback, which have significantly contributed to enhancing our understanding and improving the presentation of our work. In response to your comments, we have conducted additional analyses, clarified theoretical aspects, and provided detailed comparisons to better highlight the practical value and optimization potential of the GOOD method.
>
> We are currently incorporating these updates into the manuscript and will submit the revised version soon. We hope that the additional results and discussions provided above address your concerns and demonstrate the robustness and flexibility of our approach.
>
> If you believe our efforts have sufficiently addressed the issues raised in your review, we would greatly appreciate your reconsideration of the evaluation of our work. Please do not hesitate to let us know if there are any further questions or points requiring clarification—we would be delighted to provide additional explanations.
>
> Thank you again for your constructive feedback and the opportunity to improve our work. Your guidance has been invaluable in strengthening the quality of this research.

---

> ### Author Response · Authors · 2024-11-25
> **Response to Reviewer Yvpt's Comments and Revised Manuscript Submission**
>
> We sincerely thank you for your thoughtful and constructive feedback on our work. Your comments have been invaluable in helping us improve the quality and clarity of our work.
>
> In response to your suggestions, we have carefully revised the manuscript and submitted an updated version. Below, we summarize the key changes:
>
> 1. We have added comparisons and citations for relevant works, including DeAL, DeRa, EFT and so on.
> 2. The explanation of the GOOD algorithm has been improved, with additional details provided in the **Details of Alignment Discrimination** and **Implementation of Guidance** sections.
> 3. We updated the tables to use the three-line table format for better readability.
> 4. All figures have been checked and replaced with PDF versions to ensure they can be zoomed in for detail.
> 5. Additional discussions on the interference performance of GOOD have been included (Appendix D).
> 6. We have introduced a description of potential application scenarios for GOOD (Appendix E).
> 7. An analysis of GOOD's requirements for guiding model size has been added (Appendix F).
> 8. Analyses of the importance of **Alignment Discrimination** and **Token Substitution** are now included (Appendix G).
> 9. More comprehensive comparative results with baseline methods have been added (Appendix H).
>
> We hope these revisions address your concerns and enhance the manuscript’s overall quality and contribution.
>
> If you believe our revisions have resolved the issues raised in your review, we would greatly appreciate your reconsideration of the manuscript’s evaluation. If there are any remaining concerns or further suggestions for improvement, we would be more than happy to address them.
>
> Once again, we sincerely thank you for your time and effort in reviewing our work. Your feedback has been instrumental in shaping this improved version of our manuscript.

---

### Official Review · Reviewer_v3Pr · 2024-11-05

**Soundness:** 3
**Presentation:** 2
**Contribution:** 2
**Rating:** 3
**Confidence:** 4

**Summary:**

This paper proposes an alignment method called Guided Alignment Online Optimal Decoding (GOOD) that can enhance the outputs of a pre-trained models during response generation. This method uses a pair of guiding models, namely an aligned one and a non-aligned one, to identify critical positions and adjusts the target model’s output dynamically during decoding. The main advantage is that no finetuning or additional training required for the target model. Experiments show that this method has competitive performance compared to finetuning in terms of general capability and harmless generation,  In addition, it can be used to enhance strong non-aligned models and already aligned models.

**Strengths:**

1. Proposes a method that is able to align any large language model by having access to a pair of smaller models, a non-aligned model and its aligned version, during response generation without requiring any finetuning.
2. The key innovation that differentiates it from proxy tuning is the idea of re-encoding and using the generated tokens from the guiding model on particular locations instead of directly modifying the output distributions of the target model. This enables its application to any black-box model.
3. Extensive evaluations show that the proposed method can recover the alignment performance of finetuned models by using to weak models. Notably, this finding holds for particularly large models (improving Qwen-2-72B model using Qwen-2-7B) and it can also improve already aligned models.

**Weaknesses:**

1. Using two models in addition to the target one and performing forward passes in parallel can lead to additional computational cost and memory requirements, as well as increased complexity during deployment phase. These factors can hinder applicability and adoption in real-world scenarios. The paper could benefit from quantifying this impact.
2. The evaluation focuses mainly on comparisons with full-finetuning and lacks broader coverage of alternative alignment methods that are applied during decoding time other than Proxy Tuning and GaC. For instance, using a reward model directly during the decoding process to guide generation (https://arxiv.org/pdf/2402.06147) or through interpolation of logits (https://arxiv.org/pdf/2402.02992).
3. The necessity of using both auxiliary models haven't been justified theoretically or empirically through ablation experiments (even if it requires relaxing the blackbox assumption). I'd suggest comparing with output ensembling or speculative decoding by using the aligned auxiliary model whose output distribution you aim to maintain, to demonstrate if the proposed method is better than directly combining the aligned model with the target one.
4. Discussion is missing to highlight the differences of the proposed approach to constrained text generation literature and related methods (https://arxiv.org/pdf/2112.08726, https://arxiv.org/pdf/2104.08768).
5. There is little emphasis on the potential drawbacks and performance in the limit of the proposed method, for instance what is the minimum size of the pair of auxiliary models need to be for this approach to work?

**Questions:**

1. The introduction omits to explain the significance and impact of using a pair of guiding models in terms of cost and quality. What are the assumptions about these models? Do they require training on the same alignment data as the target model?
2. Can you elaborate on the relationship of GOOD with prior work? It would be useful to explain for the reader what are the common parts and differences compared to prior established methods for constraining the outputs of LMs.
3. What are the key advantages of GOOD compared to alternative decoding-time alignment methods such as the ones mentioned in the weakness 2 above?
4. Using three models during inference is expected to increase the latency and memory requirements dramatically. Can the authors share a few numbers that quantify the computational cost? Inference is actually the part of the LLM stack that is the most costly as it requires committed resources to it, while finetuning happens only once. If you have any comparisons between these costs it would help with better understanding the pros and cons.
5. In what kind of scenarios to the authors envision this approach to be used in practice?

---

> ### Author Response · Authors · 2024-11-24
> **Response to Reviewer v3Pr: W1, Q4**
>
> We sincerely appreciate Reviewer v3Pr's time, effort, insightful feedback, and constructive suggestions, which are valuable in helping us refine our work. Below, we address the concerns raised and provide additional discussions and justifications to clarify our contribution.
>
>
> ## W1, Q4:
>
> We conducted tests based on the current implementation of GOOD and analyzed the theoretical optimal performance of the method.
>
> ### Current Performance
>
> We evaluated the alignment of Gemma2-2b-it → Gemma2-27b and Gemma2-2b-it → Qwen2-72. The detailed test configurations are as follows:
>
> - Due to varying GPU memory requirements for different configurations, the speed measurements for Gemma2-2b-it → Gemma2-27b and Gemma2-27b-it were conducted on L40s (48GB × 8), while the speed measurements for Gemma2-2b-it → Qwen2-72 and Qwen2-72-Instruct were performed on A100 (80GB × 8).
> - For Gemma2-27b-it and Qwen2-72-Instruct, generation was conducted using the Huggingface Transformers library.
> - The test question set was sourced from MT-Bench, covering multiple question categories.
> - Model inference utilized caching, and all models involved were deployed using model parallelism.
> - We examined the decoding speed under various configurations as a function of generation length and confirmed that the number of tokens already generated had no significant impact on model inference speed.
>
> The test results show that the average decoding speed for Gemma2-2b-it → Gemma2-27b is 1.27 times that of Gemma2-27b-it, and the average decoding speed for Gemma2-2b-it → Qwen2-72 is 1.15 times that of Qwen2-72-Instruct.
>
> ### Optimization Potential
>
> It is worth noting that the current implementation of GOOD still holds significant potential for performance improvement. In the current implementation, for each token's decoding, the Guiding model pair is first inferred, and based on its judgment, it is determined whether the Guided model needs to be inferred (if alignment is deemed unnecessary, the Guided model's inference is skipped). The estimated time complexity formula for the current implementation is given below (taking the inference speed of the original model as the baseline value of 1):
>
> T(GOOD)/T(Vanilla) = (2 * α * β * 1) + (1 - Ω) + γ
>
> The formula and symbol definitions are as follows:
>
> - (2 * α * β * 1): Decoding time of the Guiding model pair (the Guiding model participates in each decoding step).
> - (1 - Ω): Decoding time of the Guided model.
> - γ: Additional time overhead caused by switching between the Guiding model pair and the Guided model for inference.
> - α: The ratio of the parameter size of a single Guiding model to that of the Guided model.
> - β: The inference speed of the Guiding model relative to the Guided model at the same parameter size.
> - Ω: The average substitution ratio of decoding by the Guiding model pair.
>
> For example, in Gemma2-2b-it → Gemma2-27b:
>
> - β = 1,
> - α = 0.074,
> - Ω = 0.3,
> - γ is estimated as 0.422.
>
> This estimation indicates that the current implementation of GOOD can further improve its speed and achieve better inference performance than Vanilla Decoding by addressing the following directions:
>
> 1. Since the two Guiding models can execute in parallel, the decoding time of the Guiding model pair could potentially be reduced from 2 * α * β * 1 to α * β * 1.
> 2. Since communication between the Guiding models involves only string exchanges with minimal overhead, the Guiding model pair and the Guided model can be deployed separately to reduce the overhead of switching models, potentially significantly decreasing γ.
> 3. Since the Guiding models already perform predictions before the Guided model's inference, the Guiding models can be viewed as Speculative Inference SSMs, with the Guided model acting as the Verifier. According to estimates from SpecInfer[1], the decoding performance of the Guided model can potentially improve by 1.3–2.4× without additional overhead.
>
> Thus, under the most ideal implementation, the GOOD decoding performance could be optimized to:
>
> (α * β * 1) + (1 - Ω) * 0.42
>
> For Gemma2-2b-it → Gemma2-27b, this value equals 0.368.
>
> Although the current implementation is far from achieving this theoretical performance, we believe that GOOD can be further improved to achieve inference performance superior to Vanilla Decoding.
>
> [1] Miao, Xupeng, et al. "Specinfer: Accelerating large language model serving with tree-based speculative inference and verification." Proceedings of the 29th ACM International Conference on Architectural Support for Programming Languages and Operating Systems, Volume 3. 2024.

---

> ### Author Response · Authors · 2024-11-24
> **Response to Reviewer v3Pr: W2, Q3**
>
> ## W2, Q3
>
> Thank you for providing these two references. We have carefully reviewed them and found that neither paper has made their code publicly available, making it difficult to replicate their functionalities within a short timeframe. At the same time, we noticed that DeAL reports its Harmless ratio on the HH Dataset. Therefore, we conducted comparative experiments using the same base model:
>
> | Method | Model | Harmless |
> | --- | --- | --- |
> | GOOD | Gemma2-2b-it→MPT-7b | 0.89 |
> | DeAL | MPT-7b | 0.67 |
>
> In this comparison, we aligned MPT-7b using Gemma2-2b-it, while DeAL utilized a fine-tuned reward model for guidance. Since this reward model has not been publicly released, we cannot evaluate its performance level, and its differences from our guiding model are significant. As a result, this comparison may lack practical significance, for which we apologize. If you have further questions or suggestions on this matter, please feel free to share them!
>
> From a theoretical perspective, the comparison is as follows: DeAL requires forward-looking inference, meaning it performs multiple inferences for each generated token, which incurs a higher cost than the overhead of GOOD's use of a pair of small models for inference. Additionally, DeAL faces challenges when aligning with multiple objectives simultaneously (e.g., "harmless" and "helpful"), whereas GOOD introduces alignment through the guiding model, which does not have this issue.
>
> As for the DeRa method mentioned in the second paper, it combines the outputs of the SFT model and the aligned model. However, determining the weighting factor for this combination is challenging, and it still requires the model to undergo basic SFT training, rendering it incapable of directly leveraging a pretrained model. Moreover, during the operation of GOOD, the guiding model pair and the guided model exchange only character-level information, allowing them to function under a bidirectional black-box setting. In contrast, DeRa requires processing the logits outputs of two models, which is a more restrictive and computationally demanding approach.

---

> ### Author Response · Authors · 2024-11-24
> **Response to Reviewer v3Pr: w3**
>
> ## W3:
>
> The GaC method compared in our paper belongs to a model ensemble approach. In the current version of the paper, we provided some comparative data. Here, we have added additional test results in the hope that they better address your concerns:
>
> - Code enhancement experiment (from the current paper)
>
>     | Method | HumanEval Pass@1 |
>     | --- | --- |
>     | CodeLlama-7b-python + Llama-2-13b-chat (GaC) | 29.9 |
>     | CodeLlama-7b-python → Llama-2-13b-chat (GOOD) | 32.1 |
>
> - Additional test results
>
>     | Method | Model | MT-Bench | AlpacaEval | Harmless |
>     | --- | --- | --- | --- | --- |
>     | GOOD | Gemma2-2b-it → Gemma2-9b | 7.81 | 32.05 | 0.95 |
>     | GaC | Gemma2-2b-it + Gemma2-9b | 5.52 | 10.12 | 0.88 |
>
> The current experimental results demonstrate that alignment guided by GOOD is significantly better than directly combining an aligned model with the target model. This indicates that employing a pair of guiding models is necessary. Our analysis in Section 5.1 of the paper also shows that accurately determining the token positions requiring alignment has a substantial impact on the performance of alignment without fine-tuning, and this discrimination should be as precise as possible.

---

> ### Author Response · Authors · 2024-11-24
> **Response to Reviewer v3Pr: W4, Q2**
>
> ## W4, Q2:
>
> Thank you very much for pointing out this issue. We will incorporate comparisons related to constrained text generation into the revised version of the paper. Here, we provide a brief theoretical comparison:
>
> - NEUROLOGIC A*esque Decoding: This method considers both the information of already generated tokens and the possible tokens to be generated in the future during decoding. It uses a heuristic approach to select the appropriate path without introducing additional models. However, considering future token information requires predictive decoding, which introduces additional computational demands.
>
> - Constrained Language Models Yield Few-Shot Semantic Parsers: This method also imposes constraints during decoding. The difference is that its constraints are based on manually defined SCFG grammars, and it prunes decoding paths according to the grammar rules without involving additional model inference. Its advantage lies in speed and strict adherence to predefined targets. However, it requires pre-defined grammar rules, making it more suitable for tasks such as code generation but less applicable to tasks like alignment.
>
> - GOOD: Compared to the constrained text generation methods mentioned above, GOOD also performs adjustments during decoding. However, instead of modifying logits or pruning paths, GOOD directly replaces decoding outputs using alternative decoding from guiding models. This characteristic of GOOD provides it with a high degree of flexibility.

---

> ### Author Response · Authors · 2024-11-24
> **Response to Reviewer v3Pr: w5**
>
> ## W5:
>
> We tested the configuration of Gemma2-2b-it → Gemma2-27b, and the MT-Bench results are as follows:
>
> | Model | MT-Bench |
> | --- | --- |
> | Gemma2-27b-it | 8.97 |
> | Gemma2-2b-it → Gemma2-27b | 8.18 |
>
> In this setup, the guiding model and the guided model differ in size by over 10 times. However, the GOOD method still achieves 91% of the performance of directly fine-tuned alignment, indicating that GOOD does not impose particularly stringent requirements on the relative size of the auxiliary models.

---

> ### Author Response · Authors · 2024-11-24
> **Response to Reviewer v3Pr: Q1**
>
> ## Q1:
>
> 1. **Guiding model pairs in GOOD**: In the GOOD method, the use of guiding model pairs is aimed at obtaining discriminative information about whether alignment is required at a specific position. Based on this information, the method decides whether to use replacement tokens from the guiding models. Therefore, with the current implementation of GOOD, the use of guiding model pairs is necessary.
>
> 2. **Model assumptions**: A fundamental assumption of using guiding models to replace tokens for the guided model is that different models exhibit similar behaviors at positions relevant to alignment.
>
> 3. **Performance analysis of GOOD**: Our analysis of GOOD indicates that its performance is influenced by two factors: the accuracy of identifying positions that require alignment and the effectiveness of the guidance provided at these positions. The latter depends on the performance of the guiding models themselves. For the former, guiding models trained on the same alignment data as the guided model typically ensure greater similarity in alignment behavior, leading to more accurate alignment identification. However, in some cases, the alignment adjustments of certain models may cover alignment-related modifications applicable to other model families. In such situations, GOOD can still function effectively even if the guiding models are not trained on the same alignment data.

---

> ### Author Response · Authors · 2024-11-24
> **Response to Reviewer v3Pr: Q5**
>
> ## Q5：
>
> We have added further explanations about the practical value of the GOOD method. We hope this provides you with a clearer understanding of the approach：
>
> ### Reducing Repetitive Fine-tuning
>
> GOOD enables fine-tuning conducted on one LLM to be transferred to another LLM, thereby avoiding unnecessary repetitive fine-tuning and reducing the number of model variants caused by different fine-tuning processes. Even if these models differ only slightly, their redundant storage can lead to significant waste of storage resources.
>
> ### Studying the Impact of Fine-tuning
>
> GOOD can be used to analyze the sources of performance gains from fine-tuning. For example, it can help determine whether the performance improvement stems from changes in linguistic habits or from deeper learning. If it is the former, simply transferring these linguistic habits to other models may achieve similar performance gains. If it is the latter, the gains from such a transfer should be significantly lower than direct fine-tuning (note that this comparison should use equally accurate alignment discrimination).
>
> ### LLM Edge-Cloud Collaboration
>
> Since GOOD involves only string-level information exchange between the guiding model pair and the guided model, it enables low-cost collaboration between edge models and cloud models during decoding. Our analysis in Section 5.2 shows that GOOD-guided alignment retains the alignment characteristics of the guiding models. This means that in LLM edge-cloud collaboration supported by GOOD, it is possible to perform customized fine-tuning of edge models without exposing user-private conversational data. This allows the collaborative output to incorporate both user-specific customization and the powerful capabilities of cloud models.
>
> In this scenario, cloud models can use not only pretrained models but also aligned models. We conducted a series of tests demonstrating that in GOOD-supported edge-cloud collaboration, overall performance improves as the cloud model’s performance enhances, even without updating the user’s edge model:
>
> | Model | MT-Bench Score |
> | --- | --- |
> | Gemma2-2b-it | 7.60 |
> | Llama3-8b-Instruct | 7.75 |
> | Gemma2-2b-it→Llama3-8b-Instruct | 7.33 |
> | Qwen2-7b-Instruct | 8.02 |
> | Gemma2-2b-it→Qwen2-7b-Instruct | 7.80 |
> | Gemma2-9b-it | 8.34 |
> | Gemma2-2b-it→Gemma2-9b-it | 8.44 |
>
> This characteristic means that GOOD can extend the lifespan of customized models by preserving the unique features of various local fine-tunings while keeping their performance up-to-date, rather than allowing them to quickly fall behind newer models and require frequent updates.
>
> ### GOOD as Further Validation of the Superficial Alignment Hypothesis
>
> The Superficial Alignment Hypothesis suggests that most of a model’s knowledge and capabilities are acquired during pretraining, with alignment primarily teaching the model which sub-distribution of responses to utilize in user interactions. The GOOD method replaces part of the decoding results of one model with alignment-related tokens from another model, without any fine-tuning, yet effectively achieves alignment. This supports the Superficial Alignment Hypothesis to some extent, indicating that alignment in models likely changes linguistic habits rather than learning new knowledge or capabilities.

---

> ### Author Response · Authors · 2024-11-24
> **Response to Reviewer v3Pr:**
>
> We sincerely appreciate Reviewer v3Pr's constructive feedback and thoughtful suggestions, which have been instrumental in helping us refine our manuscript. In response to your insightful comments, we have conducted extensive additional analyses, incorporated new experiments, and enhanced the theoretical discussions to address the raised concerns. These efforts have not only improved the clarity and depth of our work but also highlighted the robustness and flexibility of the GOOD method under diverse settings.
>
> We hope that our detailed responses, along with the additional results and analyses presented above, effectively address your concerns and demonstrate the significant potential of our approach. We are currently incorporating these revisions into the manuscript and plan to submit the updated version soon. If there are any remaining issues or further aspects you believe require clarification, we would be more than happy to address them in detail.
>
> Finally, if you find that the improvements we have made effectively resolve the issues raised in your review, we would greatly appreciate your reconsideration of the evaluation of our work. Your feedback has been invaluable to us, and we are deeply grateful for your continued guidance and support throughout this process.

---

> ### Author Response · Authors · 2024-11-25
> **Response to Reviewer v3Pr: w5 (new)**
>
> ## W5:
>
> We evaluated the performance of Qwen2.5-0.5b-Instruct guiding Qwen2-7b and Qwen2-72b, with the results shown below:
>
> | Model | MT-Bench Score |
> | --- | --- |
> | Qwen2.5-0.5b-Instruct | 5.01 |
> | Qwen2-7b | 6.65 |
> | **Qwen2.5-0.5b-Instruct → Qwen2-7b** | **6.27** |
> | Qwen2-72b | 7.63 |
> | **Qwen2.5-0.5b-Instruct → Qwen2-72b** | **7.21** |
>
> Although Qwen2.5-0.5b-Instruct can benefit from the enhancement of pretrained models when guiding Qwen2-7b and Qwen2-72b, its weaker performance compared to the guided pretrained models leads to an overall performance degradation (falling below the pretrained models' original scores). This result indicates that, at the very least, an aligned guiding model weaker than the guided pretrained model should not be used for guidance.
>
> We further evaluated the effectiveness of using Qwen2.5-0.5b-Instruct to guide Gemma2-9b, whose MT-Bench score is lower than that of the guiding model. The results show that in this case, the performance of Qwen2.5-0.5b-Instruct → Gemma2-9b surpasses both the guiding model and the guided model:
>
> | Model | MT-Bench Score |
> | --- | --- |
> | Qwen2.5-0.5b-Instruct | 5.01 |
> | Gemma2-9b | 2.69 |
> | **Qwen2.5-0.5b-Instruct → Gemma2-9b** | **5.37** |
>
> Additionally, we examined the performance of Qwen2.5-0.5b-Instruct providing guidance under the most accurate alignment discrimination (AD). In this context, "most accurate alignment discrimination" refers to comparing the aligned version of the guided model with the unaligned guided model itself (e.g., Gemma-2-9b-it and Gemma-2-9b) to determine whether alignment is needed. If alignment is required, the guiding model’s output is used at the corresponding position. The results are as follows:
>
> | Model | MT-Bench Score |
> | --- | --- |
> | Qwen2.5-0.5b-Instruct | 5.01 |
> | Gemma-2-9b | 2.69 |
> | **Qwen2.5-0.5b-Instruct → Gemma-2-9b (with AD)** | **6.70** |
> | Qwen2-7b | 6.65 |
> | **Qwen2.5-0.5b-Instruct → Qwen2-7b (with AD)** | **7.19** |
>
> These results suggest that, as long as the alignment discrimination is sufficiently accurate, even a 0.5b-parameter model can provide meaningful guidance for alignment.

---

> ### Author Response · Authors · 2024-11-25
> **Response to Reviewer v3Pr: W1, Q4 (additional)**
>
> ## W1, Q4
>
> We conducted an evaluation comparing the decoding performance of three methods: GOOD (Gemma-2-9b-it → Gemma-2-27b), Proxy-Tuning (Gemma-2-9b-it → Gemma-2-27b), and Vanilla (Gemma-2-27b-it).
>
> The first question from MT-Bench was used as the prompt, with the maximum generation length set to 512. The experiments were performed on L40s (48G × 8).
>
> The results show that the decoding speeds of GOOD and Vanilla are unaffected by generation length, whereas Proxy-Tuning’s decoding speed slows down as the generation length increases (which might be due to specific implementation details— in theory, Proxy-Tuning could also maintain a decoding speed independent of generation length).
>
> For GOOD, the decoding speed exhibits two distinct regions: one where the guided model decoding is skipped (denoted as Region A) and another where skipping does not occur (denoted as Region B). Region A demonstrates significantly faster decoding compared to Vanilla decoding. When Speculative Inference is incorporated, Region B can also be accelerated, bringing its performance closer to or even surpassing Vanilla decoding. This suggests that further improvements to GOOD are very likely to achieve overall performance superior to Vanilla decoding, not only in terms of speed but also with lower computational costs.
>
> We will include detailed charts in the revised version of the paper to be submitted soon. We hope this will make our explanation of GOOD’s inference performance clearer. Thank you for your valuable guidance!

---

> ### Author Response · Authors · 2024-11-25
> **Response to Reviewer v3Pr's Comments and Revised Manuscript Submission**
>
> We sincerely thank you for your thoughtful and constructive feedback on our work. Your comments have been invaluable in helping us improve the quality and clarity of our work.
>
> In response to your suggestions, we have carefully revised the manuscript and submitted an updated version. Below, we summarize the key changes:
>
> 1. We have added comparisons and citations for relevant works, including DeAL, DeRa, EFT and so on.
> 2. The explanation of the GOOD algorithm has been improved, with additional details provided in the **Details of Alignment Discrimination** and **Implementation of Guidance** sections.
> 3. We updated the tables to use the three-line table format for better readability.
> 4. All figures have been checked and replaced with PDF versions to ensure they can be zoomed in for detail.
> 5. Additional discussions on the interference performance of GOOD have been included (Appendix D).
> 6. We have introduced a description of potential application scenarios for GOOD (Appendix E).
> 7. An analysis of GOOD's requirements for guiding model size has been added (Appendix F).
> 8. Analyses of the importance of **Alignment Discrimination** and **Token Substitution** are now included (Appendix G).
> 9. More comprehensive comparative results with baseline methods have been added (Appendix H).
>
> We hope these revisions address your concerns and enhance the manuscript’s overall quality and contribution.
>
> If you believe our revisions have resolved the issues raised in your review, we would greatly appreciate your reconsideration of the manuscript’s evaluation. If there are any remaining concerns or further suggestions for improvement, we would be more than happy to address them.
>
> Once again, we sincerely thank you for your time and effort in reviewing our work. Your feedback has been instrumental in shaping this improved version of our manuscript.

---

> ### Comment · Reviewer_v3Pr · 2024-12-02
> **Response to Authors**
>
> Thank you for the effort to answer my questions!
>
> A few of my concerns have been addressed but I still have doubts regarding a number of them, especially, w1, w2/w4, and w5. For w1, the theoretical potential is not that useful if it is not materialized and verified in practice. For w2/4, comparing with these methods is essential in my view but I understand the time constraints; hopefully, it will be covered thoroughly in the next revision.  For w5, the experiment shared provides only a partial view of the limits of the proposed method and wasn’t very illuminating. The key issues with the present version is that the motivation/positioning with respect to prior work and head-to-head comparisons in terms of computational cost & quality with relevant prior methods are limited.
>
> For these reasons, I decided to maintain my score but I think that the paper has potential if these concerns are fully addressed in the next revision.

---

### Note · Authors · 2024-12-03

**Comment:**

Thank you for all the feedback and guidance! We will withdraw this submission and continue to improve our work.

**Withdrawal Confirmation:**

I have read and agree with the venue's withdrawal policy on behalf of myself and my co-authors.